



# Wind farm layout optimization using pseudo-gradients

Erik Quaeghebeur[1], René Bos[2], and Michiel B. Zaaijer[3]

[1]Uncertainty in AI Group, Eindhoven University of Technology, De Groene Loper 5, 5612 AZ Eindhoven, The Netherlands
[2]Eneco, Marten Meesweg 5, 3068 AV Rotterdam, The Netherlands
[3]Wind Energy Section, Delft University of Technology, Kluyverweg 1, 2629 HS Delft, The Netherlands

**Correspondence:** Michiel Zaaijer (M.B.Zaayer@tudelft.nl)

**Abstract.** This paper presents a heuristic building block for wind farm layout optimization algorithms. For each pair of wake-interacting turbines, a vector is defined. Its magnitude is proportional to the wind speed deficit of the waked turbine due to the waking turbine. Its direction is chosen from the inter-turbine, downwind, or crosswind directions. These vectors can be combined for all waking or waked turbines and averaged over the wind resource to obtain a vector, a 'pseudo-gradient', that
can take the role of gradient in classical gradient-following optimization algorithms. A proof-of-concept optimization algorithm demonstrates how such vectors can be used for computationally efficient wind farm layout optimization. Results for various sites, both idealized and realistic, illustrate the types of layout generated by the proof-of-concept algorithm. These results provide a basis for a discussion of the heuristic's strong points—speed, competitive reduction in wake losses, flexibility—and weak points—partial blindness to the objective and dependence on the starting layout. The computational speed of pseudo-
gradient-based optimization is an enabler for analyses that would otherwise be computationally impractical. Pseudo-gradient-based optimization has already been used by industry in the design of large-scale (offshore) wind farms.

**Key words:** wind farm layout, optimization, wake model, gradient, heuristic, computational efficiency

## 1 Introduction

### 1.1 Context

For any wind farm, its layout is one of the most important design choices a developer has to make. Often, the goal of a layout optimization is to obtain the lowest possible wake losses, to maximize the revenue under a fixed feed-in tariff, where a 0.1% gain in energy yield for a large wind farm can easily correspond to several million euros in revenue over its lifetime.

A layout optimization for a large (offshore) wind farm—which could involve tens to hundreds of turbines to be placed in a possibly very complex polygon—is demanding in terms of computational power since it requires a wake model run for every
cost function evaluation. For a final layout design, a runtime of several weeks is quickly justified. However, before reaching a final design, a designer usually goes through an exploratory phase where many options are still on the table, ranging from different turbine types and numbers of units to various practical constraints set by the installation contractor. Furthermore, being one of the first steps of a Levelized Cost of Energy (LCoE) assessment, layout optimizations are in general under a lot of time pressure in a real-life competitive tender process.





At the same time, the cost functions to be used in the optimizations are becoming increasingly complex. Where wake losses used to be a good gauge for the LCoE improvement, falling subsidy levels mean that the balance of plant costs play an increasingly bigger role in the layout design, which calls for an assessment of foundation weight and cable length in every cost function evaluation. Moreover, for subsidy-free wind farms, the electricity price can no longer be assumed a constant and market dynamics will have to be involved. Finally, since risk is a significant part of the LCoE assessment as well, an uncertainty evaluation method such as a stochastic simulation might also be part of the cost function, further driving up its runtime.

What makes many of the well-known optimization algorithms (e.g., genetic algorithms, particle swarm optimization) non-ideal for wind farm layout optimization is that they require many cost function evaluations for a single iteration step. In this paper, we present a new heuristic optimization algorithm that uses some of the steps of the cost function—most notably the energy losses per wind direction sector—to construct a so-called pseudo-gradient. In its simplest form, this pseudo-gradient describes the value that each wind turbine gains or loses when facing the wind from a certain direction, which can then be translated into a vector that shifts it to a new location. The major advantage of such an approach is that it only requires a single cost function evaluation for the wind farm to construct the pseudo-gradient vector for every turbine. The algorithm has been successfully used in commercial offshore wind projects.

## 1.2 Overview

This paper starts with a mathematical description of the different aspects of the wind farm layout optimization problem (Sec. 2). This establishes the concepts and mathematical formalization used in the rest of the paper. Next, pseudo-gradients themselves are defined and illustrated (Sec. 3). This answers the question of what they are in a mathematically precise way and indicates how they can form a basis for wind farm layout optimization. Then it is concretely shown how pseudo-gradients can be used for this purpose (Sec. 4.3). Namely, optimization algorithms and the results of their application to wind farm layout optimization problems are presented and discussed. Finally, the important conclusions of the research are presented and some possible lines of follow-up research are shared (Sec. 5).

## 2 Wind farm layout optimization

### 2.1 Overview

This section gives an abstract mathematical description of the models involved in the wind farm layout optimization problem. It starts with a description of the wind farm in Sec. 2.2, where the optimization problem's design variables and constraints are defined. It continues with the models that play a role in the optimization problem's objective function. Namely, those for the wind resource (Sec. 2.3), the turbine (Sec. 2.4), and the wake effects (Sec. 2.5). It closes with a description of the objective in Sec. 2.6.



The description is abstract because the approach to layout optimization presented in this paper is applicable independent of the concrete details of the models involved. For example, it can be used with a large class of wake models. This does not mean, however, that the behavior of optimization algorithms built on this approach are not affected by specific modeling choices.

## 2.2 The wind farm

For the purposes of this paper, a wind farm is fully defined by the site and the layout. The site is characterized by its surface roughness length and the location constraints turbines must satisfy, expressed abstractly as a set $\mathcal{S}$ of coordinate values. Regulations may determine a minimal inter-turbine distance $d_{\mathrm{mit}}$ characterizing the distance constraints. A location called $\sigma$ can be specified using coordinates: $\boldsymbol{\ell}_\sigma = (\boldsymbol{p}_\sigma, z_\sigma)$, where $z_\sigma$ is the height coordinate and $\boldsymbol{p}_\sigma = (x_\sigma, y_\sigma) = x_\sigma \boldsymbol{e}_{\theta_{\mathrm{ref}}} + y_\sigma \boldsymbol{e}_{\theta_{\mathrm{ref}} + \frac{\pi}{2}}$ is the planar location, with $\theta_{\mathrm{ref}}$ the reference direction for the site planar coordinate system. The set of turbines in the farm is conceptualized by a set of indices $\mathcal{T}$. So $|\mathcal{T}|$ is the number of turbines. The layout of a wind farm is then determined by the finite set of turbine hub locations $\mathcal{L} = \{\boldsymbol{\ell}_t : t \in \mathcal{T}\}$. These locations $\boldsymbol{\ell}_t$ are the design variables of the wind farm layout optimization problem.

The planar vector from turbine $t$ to turbine $\tau$ is $\boldsymbol{p}_{t \to \tau} = \boldsymbol{p}_\tau - \boldsymbol{p}_t$ and the corresponding unit vector is $\boldsymbol{e}_{t \to \tau} = \frac{\boldsymbol{p}_{t \to \tau}}{\|\boldsymbol{p}_{t \to \tau}\|}$. A layout is valid if the turbine locations satisfy the location constraints ($\mathcal{L} \subset \mathcal{S}$) and the distance constraints ($\|\boldsymbol{p}_{t \to \tau}\| \geq d_{\mathrm{mit}}$ for all distinct $t$ and $\tau$ in $\mathcal{T}$).

## 2.3 The wind resource

The wind resource at a site is mainly characterized by a joint probability distribution for wind direction $\Theta$ and free stream wind speed $U$. The joint probability distribution can be decomposed as a marginal probability distribution for the wind direction—the wind rose—and conditional probability distributions for the free stream wind speed $U^\Theta$ for a given direction. The operators $\mathbb{E}$, $\mathbb{E}_\Theta$, and $\mathbb{E}_{U^\Theta}$ denote expectation relative to the joint, marginal, and conditional probability distributions, respectively. (The expressions for concrete computation of expectations of functions of random variables can be found in App. A1.) Because of the dependence of certain variables on wind direction, it is useful to formalize downwind and (horizontal) crosswind directions as unit vectors $\boldsymbol{e}_\Theta$ and $\boldsymbol{e}_{\Theta + \frac{\pi}{2}}$, respectively.

A number of examples can clarify the use of the expectation operators and related notation:

- $\bar{u} = \mathbb{E}(U) = \mathbb{E}_\Theta(\mathbb{E}_{U^\Theta}(U^\Theta))$ is the site's expected—or mean—free stream wind speed,

- $\bar{\boldsymbol{e}}_{\bar{\theta}} = \mathbb{E}_\Theta(\boldsymbol{e}_\Theta)$ is the expectation of the downstream wind unit vector (it provides a definition of mean wind direction $\bar{\theta}$; in general $\|\bar{\boldsymbol{e}}_{\bar{\theta}}\| < 1$),

- $\bar{P}^\theta = \mathbb{E}_{U^\theta}(P(U^\theta))$ is the expected power output for a solitary turbine at the site, for wind coming from the direction $\theta$, and

- $\bar{P} = \mathbb{E}(P(U^\Theta)) = \mathbb{E}_\Theta(\mathbb{E}_{U^\Theta}(P(U^\Theta))) = \mathbb{E}_\Theta(\bar{P}^\Theta)$ is the expected power output for a solitary turbine at the site.





Random variables are denoted by uppercase letters and their values are denoted by the corresponding lowercase ones. (Due to convention and practicality, some non-random variables and parameters, such as $D$ and $P$, are also denoted by an uppercase letter.) Expected values—means—of random variables get a bar on top, which is also used for expectations of functions of random variables.

Next to the wind direction and speed distributions, the wind resource includes constants describing further atmospheric conditions, such as turbulence intensity. Also, the wind resource depends on the height above the surface: there is vertical wind shear. A site's wind resource is normally available at a single reference height, but it is needed at other heights, namely, hub height and possible other heights of points on the rotor disc. The dependence on height is formalized using logarithmic and power law profiles, parametrized by the roughness length. We can assume that we have a site-specific function that maps

speeds at reference height to any given height. In this paper this is not made explicit, given that it has no relevant effect for our application, but it is implicitly assumed to be applied as needed.

### 2.4   The turbine

For the purposed of this paper, a wind turbine is fully characterized by its hub height, rotor diameter $D$, power curve $P$, which maps wind speed at hub height to turbine power output, and thrust curve, which maps wind speed at hub height to the turbine

thrust coefficient. The power curve and thrust curve are usually provided as tables of values for a discrete set of wind speeds, but by interpolation a power or thrust coefficient value can be obtained for any wind speed.

    We only consider farms with a single turbine type and with a constant hub height. The approach presented in this paper is essentially unaffected if these assumptions are relaxed.

### 2.5   The wake effects

#### 2.5.1   The wake function

A wind turbine in operation affects the wind in its vicinity. Important for wind farms is the mid-to-far wake downstream of a turbine, because it is a region with decreased wind speeds, resulting in lower power production of turbines located in the wake. High-fidelity modeling—using computational fluid dynamics—of wakes and their interaction in a wind farm is too computationally demanding for wind farm layout optimization purposes. Therefore, simpler engineering wake models are used,

such as those proposed by Katić et al. (1987, 'Jensen's model') and Bastankhah and Porté-Agel (2014, the 'EPFL model'). The papers by Archer et al. (2018) and Polster et al. (2018) provide recent reviews of such models.

    For the purposes of this paper, we only need a high-level characterization of such engineering wake models. Namely, we use a function $w$ that maps the representative inflow wind speed $U_t^\Theta$ at the wake-generating turbine $t$ to a 'wake' wind speed at any location $\sigma$ in the region covered by the wake model, taking into account the replenishing effect of the surrounding free stream

wind $U^\Theta$. Locations outside this region are assumed to be unaffected. For the wake function, the wind direction-dependent





downwind and crosswind distances from $t$ to $\sigma$ are required:

$$x^\Theta_{t\to\sigma} = \boldsymbol{p}_{t\to\sigma} \cdot \boldsymbol{e}_\Theta \quad \text{(downwind)},$$

$$y^\Theta_{t\to\sigma} = \boldsymbol{p}_{t\to\sigma} \cdot \boldsymbol{e}_{\Theta+\frac{\pi}{2}} \quad \text{(horizontal crosswind)}, \qquad\qquad z_{t\to\sigma} = z_\sigma - z_t \quad \text{(vertical crosswind)},$$

where $\boldsymbol{p}_{t\to\sigma} = \boldsymbol{p}_\sigma - \boldsymbol{p}_t$ and '$\cdot$' denotes the scalar product. Gathered in a tuple, we have $\boldsymbol{\ell}^\Theta_{t\to\sigma} = (\boldsymbol{p}^\Theta_{t\to\sigma}, z_{t\to\sigma})$ with $\boldsymbol{p}^\Theta_{t\to\sigma} =$

$(x^\Theta_{t\to\sigma}, y^\Theta_{t\to\sigma})$. Then the waked wind speed can be written compactly as $U^\Theta_{\sigma\leftarrow t} = w(U^\Theta, U^\Theta_t, \boldsymbol{\ell}^\Theta_{t\to\sigma})$. The wake function expression may of course include environmental parameters such as turbulence intensity and wind speed-dependent values such as the turbine's thrust coefficient, but we can leave those implicit.

### 2.5.2   Rotor disc averaging

Points on the rotor disc of a turbine $\tau$ that finds itself in the region covered by the wake model are of course those of interest
for their effect on its power output. The vector from waking turbine hub to waked turbine hub is $\boldsymbol{\ell}^\Theta_{t\to\tau}$. Then the vector to any point $\sigma$ on the rotor disc can be written as $\boldsymbol{\ell}^\Theta_{t\to\sigma} = \boldsymbol{\ell}^\Theta_{t\to\tau} + \boldsymbol{r}$, where $\boldsymbol{r}$ is a vector from the hub to the rotor disc point. (In aligned flow, $\boldsymbol{r}$ will be a crosswind vector, but conditions like yaw misalignment lead to an additional orthogonal component.) The wake wind speed at this point is then $U^\Theta_{\tau, \boldsymbol{r}\leftarrow t} = w(U^\Theta_t, U^\Theta, \boldsymbol{\ell}^\Theta_{t\to\tau} + \boldsymbol{r})$. Often a set of rotor disc points will be of interest, which corresponds to a set of vectors $\{\boldsymbol{\ell}^\Theta_{t\to\tau} + \boldsymbol{r} : \boldsymbol{r} \in \mathcal{R}\}$. Applying the wake function then results in the wake wind field over
the waked rotor disc:

$$\mathcal{U}^\Theta_{\tau\leftarrow t} = \mathcal{W}(U^\Theta_t, U^\Theta, \boldsymbol{\ell}^\Theta_{t\to\tau}, \mathcal{R}) = \big\{ (\boldsymbol{r}, U^\Theta_{\tau, \boldsymbol{r}\leftarrow t}) : \boldsymbol{r} \in \mathcal{R} \big\},$$

where the function $\mathcal{W}$ generalizes $w$ to a set of rotor disc points.

Wakes are one reason why there can be a non-constant inflow wind speed over the rotor disc of any turbine $t$. Wind shear is another. So irrespective of its origins, we can consider a wind field $\mathcal{U}_t$ over the rotor disc, or, more precisely, the points defined
by $\mathcal{R}$. Engineering wake models and the power curve take a single, representative wind speed as an argument. So we need an averaging function $a$ that takes the wind field as an argument and returns the representative wind speed: $U_t = a(\mathcal{U}_t)$. In case $\mathcal{R}$ just consists of the hub, this function is normally taken to be trivial: $a\big(\{(\boldsymbol{0}, U_{t,\boldsymbol{0}})\}\big) = U_t$. For $\mathcal{R}$ containing a finite number of points, $a$ must be some quadrature rule. An example where $\mathcal{R}$ consists of the continuum of all rotor disc points occurs with Jensen's model, where a piecewise constant function must be integrated over the rotor disc to calculate $a$ (cf., e.g., Feng and
Shen, 2015b, Sect. 2.2).

### 2.5.3   Wake mixing

In a wind farm, a turbine $\tau$ is in general exposed to the effect from multiple waking turbines, gathered in the set $\mathcal{T}^\Theta_{\tau\leftarrow}$. Therefore, a function $c$ is needed that models the mixing (combination) of individual wakes. Consider a point $\sigma$ on the rotor disc. The function $c$ must return a combined-wake wind speed for a given free stream wind speed $U^\Theta$ and a given set $\{U^\Theta_{\sigma\leftarrow t} : t \in \mathcal{T}^\Theta_{\tau\leftarrow}\}$
of single-wake wind speeds as inputs:

$$U^\Theta_\sigma = c\big(U^\Theta, \{U^\Theta_{\sigma\leftarrow t} : t \in \mathcal{T}^\Theta_{\tau\leftarrow}\}\big).$$



Usually, this combination function is based on the root-sum-square of wind speed deficits (Katić et al., 1987). Namely, let

$$\Delta^\Theta_{\sigma \leftarrow t} = 1 - \frac{U^\Theta_{\sigma \leftarrow t}}{U^\Theta}$$

be the deficit for the point $\sigma$ due to turbine $t$ in isolation. Then

$$\Delta^\Theta_\sigma = \sqrt{\sum_{t \in \mathcal{T}^\Theta_{\tau \leftarrow}} (\Delta^\Theta_{\sigma \leftarrow t})^2}$$

is its root-sum-square combination. The combined-wake wind speed is then defined as

$$U^\Theta_\sigma = \big(1 - \varphi(\Delta^\Theta_\sigma)\big)U^\Theta.$$

Here, $\varphi$ is some saturating function, included to avoid negative or also zero wind speeds. It could be, for example, $\min\{1, \cdot\}$, $\tanh$, or $\cdot/\sqrt{1 + \cdot^2}$.

### 2.5.4   Blame fractions

In principle the combination function needs to be applied before the averaging function to obtain a representative inflow wind speed, so

$$U^\Theta_\tau = a(\mathcal{U}^\Theta_\tau) = a\bigg(\Big\{\big(\boldsymbol{r}, U^\Theta_{\tau,\boldsymbol{r}}\big) : \boldsymbol{r} \in \mathcal{R}\Big\}\bigg) = a\bigg(\Big\{\big(\boldsymbol{r}, c(U^\Theta, \{U^\Theta_{\tau,\boldsymbol{r} \leftarrow t} : t \in \mathcal{T}^\Theta_{\tau \leftarrow}\})\big) : \boldsymbol{r} \in \mathcal{R}\Big\}\bigg).$$

However, to simplify calculations, it is often done the other way around (see, e.g., Feng and Shen, 2015b, Eq. 7), so

$$U^\Theta_\tau = c\big(U^\Theta, \{U^\Theta_{\tau \leftarrow t} : t \in \mathcal{T}^\Theta_{\tau \leftarrow}\}\big) = c\Big(U^\Theta, \{a(\mathcal{U}^\Theta_{\tau \leftarrow t}) : t \in \mathcal{T}^\Theta_{\tau \leftarrow}\}\Big).$$

In whatever way this is done and which precise functions $a$ and $c$ are chosen matter for the purposes for this paper only because of the fact that it determines whether or not a precise fraction $\Lambda^\Theta_{\tau \leftarrow t}$ of the total deficit $\Delta^\Theta_\tau$ can be blamed on each of the waking turbines $t$. These fractions will be used as weights in the definition of pseudo-gradients, characterizing the relative impact of each waking turbine (cf. Sec. 3.3).

For root-sum-square deficit combination done after averaging, it is straightforward to calculate these blame fractions:

$$\Lambda^\Theta_{\tau \leftarrow t} = \frac{(\Delta^\Theta_{\tau \leftarrow t})^2}{(\Delta^\Theta_\tau)^2}.$$

If averaging is done after combination, one would also need to average blame fractions, making things substantially more involved.

The rest of this paper ignores the order in which $a$ and $c$ are applied by considering both the cases where blame fractions can or cannot be (practically) defined. (The latter case also includes models where no separate wake and combination function can be distinguished, e.g., based on computational fluid dynamics.) So we will instead use a function $b$ that summarizes the effects of both $a$ and $c$:

$$U^\Theta_\tau = b\big(U^\Theta, \{\mathcal{U}^\Theta_{\tau \leftarrow t} : t \in \mathcal{T}^\Theta_{\tau \leftarrow}\}\big).$$

Some computational considerations on wake wind speed calculations are discussed in App. B1.





## 2.6 The objective

The objective we consider here is the normalized expected farm wake loss; it must be minimized and is therefore also called the cost function. The loss is in terms of energy (or power) production. The expectation is taken over the wind resource (cf. Sec. 2.3). The normalization is relative to the hypothetical case without wakes. This objective is formalized below.

A solitary turbine at the site, so without wakes, would produce a power $P(U^{\Theta})$, with expectation $\bar{P} = \mathbb{E}(P(U^{\Theta}))$. In this paper, this value is the same for turbines, as these are assumed to be identical. In the waked case, each turbine $\tau$ produces a power $P(U_{\tau}^{\Theta})$ with expectation $\bar{P}_{\tau} = \mathbb{E}(P(U_{\tau}^{\Theta}))$; these may differ from the production of others. The turbine wake loss is $L_{\tau}^{\Theta} = P(U^{\Theta}) - P(U_{\tau}^{\Theta})$. Its expectation is $\bar{L}_{\tau} = \mathbb{E}(L_{\tau}^{\Theta}) = \bar{P} - \bar{P}_{\tau}$. This is normalized by dividing by the expected wakeless turbine power $\bar{P}$, so

$$\frac{\bar{L}_{\tau}}{\bar{P}} = 1 - \frac{\bar{P}_{\tau}}{\bar{P}}.$$

The corresponding farm-level quantities are obtained by considering all turbines. The farm wake loss is $L^{\Theta} = \sum_{\tau \in \mathcal{T}} L_{\tau}^{\Theta}$. Its expectation is $\bar{L} = \mathbb{E}(L^{\Theta}) = \sum_{\tau \in \mathcal{T}} \mathbb{E}(L_{\tau}^{\Theta}) = \sum_{\tau \in \mathcal{T}} \bar{L}_{\tau}$. This is now normalized by dividing by the expected wakeless farm power, so

$$\frac{\bar{L}}{|\mathcal{T}|\bar{P}} = \frac{1}{|\mathcal{T}|} \sum_{\tau \in \mathcal{T}} \frac{\bar{L}_{\tau}}{\bar{P}} = 1 - \frac{\sum_{\tau \in \mathcal{T}} \bar{P}_{\tau}}{|\mathcal{T}|\bar{P}}.$$

where the last equality shows that the normalized expected farm wake loss is the same as the mean normalized expected turbine wake loss. (This holds because all turbines are assumed to be identical.) The quantity $\frac{\bar{L}}{|\mathcal{T}|\bar{P}}$ is the objective.

    Minimizing the objective considered is equivalent to maximizing (expected) annual energy production (AEP), which is proportional to $\sum_{\tau \in \mathcal{T}} \bar{P}_{\tau}$. However, for presentation purposes, normalized expected farm (wake) loss has advantages. As a relative quantity, it facilitates comparing different layouts and even inter-site comparisons. AEP as an absolute, wind farm

nameplate capacity-specific quantity makes this difficult, as the bounding wakeless reference value is not immediately apparent. Of course AEP can be replaced by normalized expected farm (wake) yield or farm efficiency. Still, wake losses are generally small relative to yields, and smaller numbers are easier to digest (e.g., 4.2%–5.1% vs. 94.9%–95.8%).

## 3 Pseudo-gradients

### 3.1 Overview

This section introduces the so-called pseudo-gradient vectors that can form the basis for heuristic wind farm layout optimization. The definition of the pseudo-gradients is built-up step by step. First, only a single wind case (i.e., a single wind direction and wind speed) and a single wake interaction is considered (Sec. 3.2). Then, a single wind case is combined with multiple wake interactions (Sec. 3.3). Next, multiple wind cases are combined with a single wake interaction (Sec. 3.4). Finally, full generality is reached when multiple wind cases and multiple wake interactions are combined (Sec. 3.5).

There are multiple types of pseudo-gradients that we propose. In every step each of these types is discussed.





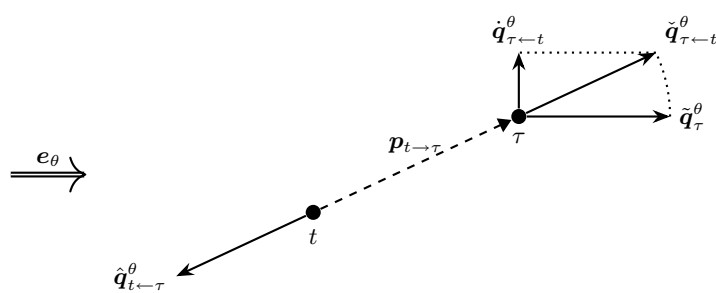

**Figure 1.** Illustration of the four pseudo-gradient vectors for the single wind case, single wake interaction case.

### 3.2 Single wind case, single wake interaction

First consider just a pair of turbines $t$ and $\tau$, a single wind direction $\Theta = \theta$, and a single wind speed $U^\theta = u$. Figure 1 shows this setup, including the four pseudo-gradient vectors defined below, where it is also discussed. In terms of wind speed, the effect waking turbine $t$ on waked turbine $\tau$ is

$$u_\tau = b(u, \{\mathcal{U}_{\tau \leftarrow t}\}) = b\Big(u, \{\mathcal{W}(u, u, \boldsymbol{\ell}^\theta_{t \to \tau}, \mathcal{R})\}\Big). \tag{1}$$

(Because there is only one waking turbine, $u_\tau$ is actually also equal to $a\big(\mathcal{W}(u, u, \boldsymbol{\ell}^\theta_{t \to \tau}, \mathcal{R})\big)$, but this simplification is of no use further on.) The effect in terms of power loss is $L^\theta_\tau = P(u) - P(u_\tau)$.

The wake power loss combines with the wind direction into what we call the simple pseudo-gradient vector:

$$\tilde{\boldsymbol{q}}^\theta_\tau = L^\theta_\tau \boldsymbol{e}_\theta. \tag{2}$$

It points downstream with a magnitude equal to the wake power loss. A spatial vector can be derived from it by multiplying it with some proportionality constant. Moving the waked turbine $\tau$ according to this vector, would place it further downstream from the wake-generating turbine $t$. This reduces the wake effect and therefore the resulting wake power loss. This may seem trivial, but forms the basic principle of optimization using pseudo-gradients. Figure 1 provides an illustration. For the purpose of clarity, the wake effect has been exaggerated for the given angle between wind direction and inter-turbine vector. (This will also be the case for further such illustrations.) For a single given wind direction $\theta$ and pair of waking turbine $t$ and waked turbine $\tau$, it shows the vector attached to the waked turbine as it would be used for layout optimization purposes.

A next type of pseudo-gradient follows from combining the wake power loss with the unit vector that points from the waking turbine $t$ to the waked turbine $\tau$:

$$\check{\boldsymbol{q}}^\theta_{\tau \leftarrow t} = L^\theta_\tau \boldsymbol{e}_{t \to \tau}. \tag{3}$$

We call it a push-away pseudo-gradient vector. Again, after converting it to a spatial vector, it can be used to move the waked turbine away from the waking turbine, reducing the wake power loss. Figure 1 shows the vector attached to the waked turbine. The dotted line between it and the simple pseudo-gradient-derived vector illustrates that for this single wind case they only differ in orientation, not magnitude.





Instead of moving the waked turbine away, it is also possible to move the waking turbine back. This idea can be implemented using what we call a push-back pseudo-gradient vector:

$$\hat{\boldsymbol{q}}^{\theta}_{t\leftarrow\tau} = L^{\theta}_{\tau}\boldsymbol{e}_{\tau\rightarrow t}. \tag{4}$$

Attached to the waking turbine and converted to a spatial vector, it moves the waking turbine away from the waked one. It
has the same effect in terms of wake power loss reduction as the corresponding push-away vector. Figure 1 shows this vector, attached now to the waking turbine.

A final type is derived from push-away vectors, by considering their projection on the crosswind direction:

$$\dot{\boldsymbol{q}}^{\theta}_{\tau\leftarrow t} = (\check{\boldsymbol{q}}^{\theta}_{\tau\leftarrow t} \cdot \boldsymbol{e}_{\theta+\frac{\pi}{2}})\boldsymbol{e}_{\theta+\frac{\pi}{2}} = L^{\theta}_{\tau}(\boldsymbol{e}_{t\rightarrow\tau} \cdot \boldsymbol{e}_{\theta+\frac{\pi}{2}})\boldsymbol{e}_{\theta+\frac{\pi}{2}} = L^{\theta}_{\tau}\frac{y^{\theta}_{t\rightarrow\tau}}{\|\boldsymbol{p}_{t\rightarrow\tau}\|}\boldsymbol{e}_{\theta+\frac{\pi}{2}}. \tag{5}$$

We call it the push-cross pseudo-gradient vector. A corresponding spatial vector attached to the waked turbine moves it away
from the center line of the wake, in that way reducing the wake effects and the wake power loss. It can be seen as a wake evasion strategy. Figure 1 shows the vector attached again to the waked turbine. The dotted line between it and the push-away pseudo-gradient-derived vector illustrates that $\dot{\boldsymbol{q}}^{\theta}_{\tau\leftarrow t}$ is the projection of $\check{\boldsymbol{q}}^{\theta}_{\tau\leftarrow t}$ on the cross-wind direction $\theta + \frac{\pi}{2}$.

One can conceive more types of pseudo-gradients than the four presented here. For example, by projecting the push-back pseudo-gradient vector on the crosswind direction, a second push-cross type vector can be defined. Systematizing, there are
three choices to make:

– associated (attached) to the waking or the waked turbine;

– oriented along the downwind direction, crosswind direction, inter-turbine direction, or the direction orthogonal to the inter-turbine one;

– defined directly or by projection.

For the four presented pseudo-gradients, we have

– simple: waked, downwind, direct;

– push-away: waked, inter-turbine, direct;

– push-back: waking, inter-turbine, direct;

– push-cross: waked, crosswind, projected.

These four presented pseudo-gradients already provide sufficient variation for this seminal investigation of pseudo-gradients for layout optimization. However, that does not imply that other variants cannot be useful in such a context. Nevertheless, some combinations are more natural: direct inter-turbine for distancing turbines and projected crosswind for wake evasion. The direction orthogonal to the inter-turbine one seems fit for neither purpose. The simple pseudo-gradients can be seen as a poor man's push-away pseudo-gradient when calculating blame fractions is impractical.



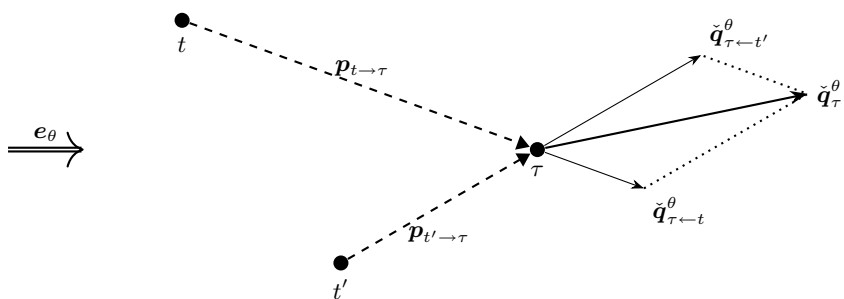

**Figure 2.** Illustration of how turbine-specific push-away pseudo-gradient vectors combine into a combined push-away pseudo-gradient vector.

### 3.3 Single wind case, multiple wake interactions

Again consider a single wind direction $\Theta = \theta$ and a single wind speed $U^\theta = u$. But now consider multiple waking or waked turbines. The expression for the waked wind speed is unchanged from before (cf. Eq. 1), as is the one for the power loss $L_\tau^\theta$. However, now there are possibly multiple wake interactions causing this loss. In case blame fractions can be calculated (cf. Sec. 2.5.4), these can be used to divide the power loss over the single turbine-to-turbine interactions involved:

$$L_{\tau \leftarrow t}^\theta = \lambda_{\tau \leftarrow t}^\theta L_\tau^\theta \tag{6}$$

and so by definition of blame fractions, we have that

$$\sum_{t \in \mathcal{T}_{\tau \leftarrow}^\theta} L_{\tau \leftarrow t}^\theta = L_\tau^\theta. \tag{7}$$

Simple pseudo-gradient vectors are aligned with the wind direction. Therefore its defining expression, Eq. 2, is unchanged, because Eq. 7 causes any decomposition into fragments $L_{\tau \leftarrow t}^\theta e_\theta$ to recombine into $L_\tau^\theta e_\theta$. (This would not hold for a waking variant; see the discussion for push-back vectors below for a similar difference.) This independence of blame fractions is what makes simple pseudo-gradient vectors applicable even if those blame fractions cannot be calculated, in contrast to the other pseudo-gradients we discuss.

The push-away pseudo-gradient vector for the case of multiple waking turbines is defined by summing over those for single wake interactions (cf. Eq. 3):

$$\check{q}_\tau^\theta = \sum_{t \in \mathcal{T}_{\tau \leftarrow}^\theta} \check{q}_{\tau \leftarrow t}^\theta = \sum_{t \in \mathcal{T}_{\tau \leftarrow}^\theta} L_{\tau \leftarrow t}^\theta e_{t \rightarrow \tau}. \tag{8}$$

This sum is illustrated in Fig. 2 for two waking turbines $t$ and $t'$ and one waked turbine $\tau$. One can see that the corresponding planar vector will move the waked turbine, relatively speaking, farther away from the waking turbine that is most to blame for the power loss.

The combined push-cross pseudo-gradient vector is closely related to the combined push-away pseudo-gradient vector. As before, it is its projection on the crosswind direction, or, equivalently because of the linearity of the projection operation, the



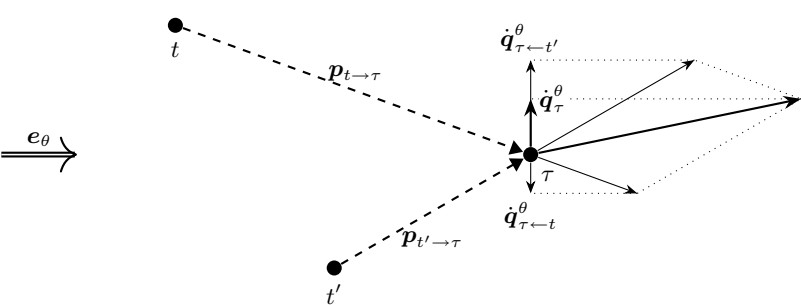

**Figure 3.** Illustration of how turbine-specific push-cross pseudo-gradient vectors combine into a combined push-cross pseudo-gradient vector and their relation by projection to push-away pseudo-gradient vectors.

sum of the push-cross vectors for single wake interactions (cf. Eq. 5):

$$\dot{\boldsymbol{q}}_\tau^\theta = (\check{\boldsymbol{q}}_\tau^\theta \cdot \boldsymbol{e}_{\theta+\frac{\pi}{2}})\boldsymbol{e}_{\theta+\frac{\pi}{2}} = \sum_{t \in \mathcal{T}_{\tau\leftarrow}^\theta} (\check{\boldsymbol{q}}_{\tau\leftarrow t}^\theta \cdot \boldsymbol{e}_{\theta+\frac{\pi}{2}})\boldsymbol{e}_{\theta+\frac{\pi}{2}} = \sum_{t \in \mathcal{T}_{\tau\leftarrow}^\theta} \dot{\boldsymbol{q}}_{\tau\leftarrow t}^\theta = \sum_{t \in \mathcal{T}_{\tau\leftarrow}^\theta} L_{\tau\leftarrow t}^\theta \frac{y_{t\rightarrow\tau}^\theta}{\|\boldsymbol{p}_{t\rightarrow\tau}\|} \boldsymbol{e}_{\theta+\frac{\pi}{2}}. \tag{9}$$

This sum and the projections are illustrated in Fig. 3 for the same turbines $t$, $t'$, and $\tau$ as in Fig. 2. In this illustration, one can see that in the definition of (combined) push-cross pseudo-gradient vectors effectively a side must be chosen. The effect is

that the waked turbine moves away from the turbines responsible to the largest part of the wake power losses, but closer to the others. So there is a qualitatively different effect compared to the other pseudo-gradient types treated; a real trade-off is made. Quantitatively, there is also a difference, as the magnitude of the combined push-cross vector is substantially smaller than the push-away one. This is due to the projection and the summing of vectors of opposing orientation.

The combined push-back pseudo-gradient vector arises differently from the push-away one, because now we must sum over

vectors for waked turbines instead of those for waking turbines. But apart from that, things are the same; namely, we again must sum over push-away vectors for single wake interactions (cf. Eq. 4):

$$\hat{\boldsymbol{q}}_t^\theta = \sum_{\tau \in \mathcal{T}_{t\rightarrow}^\theta} \hat{\boldsymbol{q}}_{t\leftarrow\tau}^\theta = \sum_{\tau \in \mathcal{T}_{t\rightarrow}^\theta} L_{\tau\leftarrow t}^\theta \boldsymbol{e}_{\tau\rightarrow t} \tag{10}$$

This sum is illustrated in Fig. 4 for one waking turbine $t$ and two waked turbines $\tau$ and $\tau'$. One can see that the corresponding planar vector will move the waking turbine, relatively speaking, farther away from the waked turbine that it affects the most in

terms of power loss.

### 3.4   Multiple wind cases, single wake interaction

Return to the two-turbine setup of Sec. 3.2. But now consider multiple wind cases, or, in mathematical terms, random variables for wind direction ($\Theta$ instead of $\theta$) and wind speed ($U^\Theta$ instead of $u$). So expectations over these random variables of the pseudo-gradient vectors defined in Eqs. 2 to 5 must be considered. As before, the waking turbine is denoted by $t$ and the waked

turbine by $\tau$



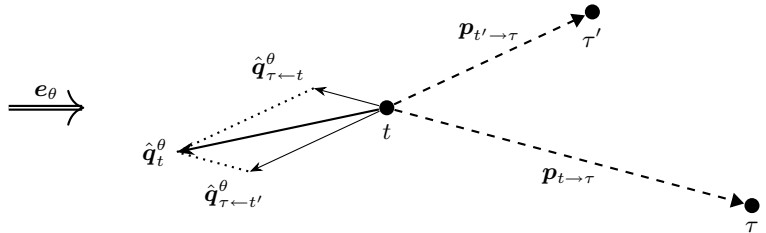

**Figure 4.** Illustration of how turbine-specific push-back pseudo-gradient vectors combine into a combined push-back pseudo-gradient vector.

Common in these defining expressions is the appearance of the wake power loss $L_\tau^\Theta$. It is the only factor in these expressions that depends on $U^\Theta$. Therefore, we can develop the impact of expectation over $U^\Theta$ in a uniform way. Let $g$ be a function that may depend on wind direction $\Theta$ and possibly other variables $o$, then (cf. Sec. 2.3)

$$\mathbb{E}\big(L_\tau^\Theta g(\Theta, o)\big) = \mathbb{E}_\Theta\big(\mathbb{E}_{U^\Theta}(L_\tau^\Theta)g(\Theta, o)\big) = \mathbb{E}_\Theta\big(\bar{L}_\tau^\Theta g(\Theta, o)\big), \tag{11}$$

where $\bar{L}_\tau^\Theta = \mathbb{E}_{U^\Theta}(L_\tau^\Theta)$. In case $g$ does not depend on $\Theta$, we get

$$\mathbb{E}\big(L_\tau^\Theta g(o)\big) = \mathbb{E}_\Theta\big(\bar{L}_\tau^\Theta\big)g(o) = \bar{L}_\tau g(o), \tag{12}$$

where $\bar{L}_\tau = \mathbb{E}(\bar{L}_\tau^\Theta)$ as in Sec. 2.6.

Applying the expectation to the expression of Eq. 2 for the simple pseudo-gradient vector gives

$$\tilde{\boldsymbol{q}}_\tau = \mathbb{E}(\tilde{\boldsymbol{q}}_\tau^\Theta) = \mathbb{E}(L_\tau^\Theta \boldsymbol{e}_\Theta) = \mathbb{E}_\Theta(\bar{L}_\tau^\Theta \boldsymbol{e}_\Theta). \tag{13}$$

This expectation is illustrated in Fig. 5 for two wind directions, $\boldsymbol{e}_\theta$ and $\boldsymbol{e}_{\theta'}$ and a single wind speed. (For the multiple-speed case the same picture would apply, with $\tilde{\boldsymbol{q}}_\tau^\theta = L_\tau^\theta \boldsymbol{e}_\theta$ and $\tilde{\boldsymbol{q}}_\tau^{\theta'} = L_\tau^{\theta'} \boldsymbol{e}_{\theta'}$ replaced by $\bar{L}_\tau^\theta \boldsymbol{e}_\theta$ and $\bar{L}_\tau^{\theta'} \boldsymbol{e}_{\theta'}$, respectively. Since the downwind unit vector is a function of wind direction only, the expectation of the loss for a certain wind direction can be separated according to Eq. 11. The same argument can be made for the illustrations for the other types of pseudo-gradients shown below.) One can see that the per-direction pseudo-gradient vectors have different directions and so a non-trivial vector

average is taken. While the direction aligned most with the inter-turbine vector results in the largest per-direction pseudo-gradient vector (here $\|\tilde{\boldsymbol{q}}_\tau^{\theta'}\| > \|\tilde{\boldsymbol{q}}_\tau^\theta\|$), its impact on the expectation is modulated by the relative weight of the wind directions in the wind rose (here $\theta$ is more probable than $\theta'$).

This is not the case for push-away pseudo-gradient vectors. Applying the expectation to the expression of Eq. 3 gives

$$\check{\boldsymbol{q}}_{\tau\leftarrow t} = \mathbb{E}(\check{\boldsymbol{q}}_{\tau\leftarrow t}^\Theta) = \mathbb{E}(L_\tau^\Theta \boldsymbol{e}_{t\to\tau}) = \bar{L}_\tau \boldsymbol{e}_{t\to\tau}. \tag{14}$$

This expression shows that, because only the single direction $\boldsymbol{e}_{t\to\tau}$ independent of the wind direction is used, the result is effectively obtained as a scalar average of losses. This expectation is illustrated in Fig. 6, again for two wind directions, $\boldsymbol{e}_\theta$ and $\boldsymbol{e}_{\theta'}$ and a single wind speed.



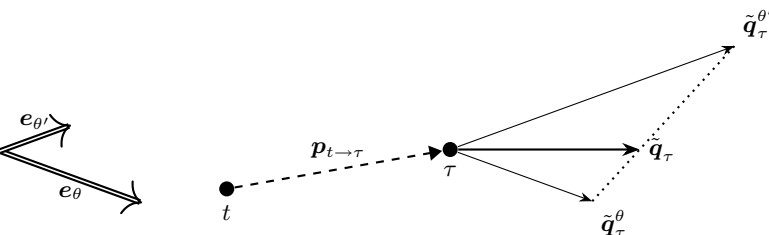

**Figure 5.** Illustration of how wind direction-specific simple pseudo-gradient vectors combine into an averaged simple pseudo-gradient vector.

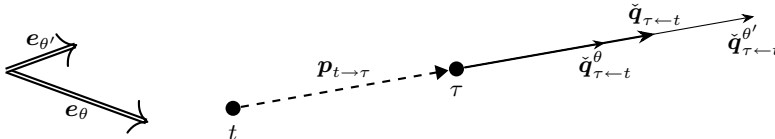

**Figure 6.** Illustration of how wind direction-specific push-away pseudo-gradient vectors combine into an averaged push-away pseudo-gradient vector.

Push-back pseudo-gradient vectors behave similarly. Applying the expectation to the expression of Eq. 4 gives

$$\hat{q}_{t\leftarrow\tau} = \mathbb{E}(\hat{q}_{t\leftarrow\tau}^{\Theta}) = \mathbb{E}(L_\tau^{\Theta} e_{\tau\rightarrow t}) = \bar{L}_\tau e_{\tau\rightarrow t} \tag{15}$$

The only difference with the push-away vector is the sense of the vector. This expectation is illustrated in Fig. 7 for the same setup as above.

5    Things become interesting again for the push-cross pseudo-gradient vectors. Applying the expectation to the expression of Eq. 5 now gives

$$\dot{q}_{\tau\leftarrow t} = \mathbb{E}(\dot{q}_{\tau\leftarrow t}^{\Theta}) = \mathbb{E}\left(L_\tau^{\Theta} \frac{y_{t\rightarrow\tau}^{\Theta}}{\|p_{t\rightarrow\tau}\|} e_{\Theta+\frac{\pi}{2}}\right) = \mathbb{E}_\Theta\left(\bar{L}_\tau^{\Theta} \frac{y_{t\rightarrow\tau}^{\Theta}}{\|p_{t\rightarrow\tau}\|} e_{\Theta+\frac{\pi}{2}}\right). \tag{16}$$

As was the case for simple pseudo-gradients, the expectation cannot be worked out completely and the unit vector is direction-dependent, so that a non-trivial vector average results. This is visible in Fig. 8, which reminds us that the direction-dependence

10   stems from the projection used for this type of pseudo-gradient. An effect of this projection also visible in the figure is that these

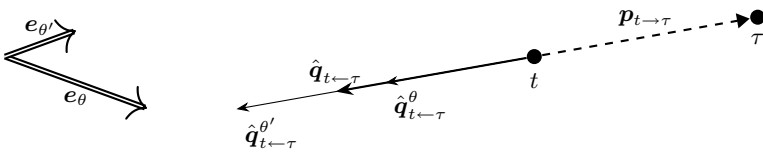

**Figure 7.** Illustration of how wind direction-specific push-back pseudo-gradient vectors combine into an averaged push-back pseudo-gradient vector.





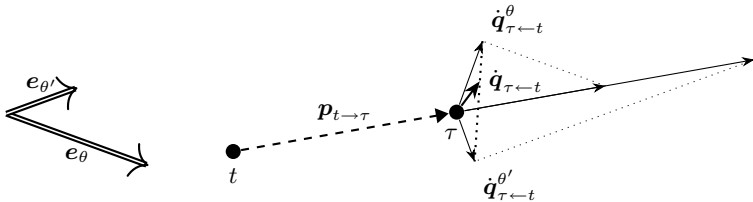

**Figure 8.** Illustration of how wind direction-specific push-cross pseudo-gradient vectors combine into an averaged push-cross pseudo-gradient vector.

pseudo-gradient vectors are considerably smaller in magnitude than the push-away pseudo-gradient vectors they are derived from.

The illustrations of Figs. 5–8 use just two wind directions, which are moreover not very different. In reality, all directions must generally be taken into account. For many of these directions, the wake effect is small or even non-existent, resulting in
pseudo-vectors of negligible magnitude. The effect is that in general after averaging the magnitude of the resulting pseudo-vectors is significantly reduced relative to the largest wind direction-specific ones.

### 3.5    Multiple wind cases, multiple wake interactions

To define pseudo-gradients for the fully general case requires considering both multiple wind cases and multiple wake interactions. Multiple wake cases are described by taking a finite sum of simple single-wake cases (see Sec. 3.3). The terms
appearing in this sum depend on the wind direction, as can be seen in Eqs. 8–10. An expectation operation describes the effect of multiple wind cases (see Sec. 3.4). Considering both can be done by applying the expectation after the summation. However, when implicitly setting (undefined) blame fractions for non-waking turbines to zero, the sum becomes independent of the wind direction. Then the order can be switched, because of the linearity of the expectation operation and the finite nature of the wake interaction sum.

For the simple pseudo-gradient vector, the argument made in Sec. 3.3 holds (no blame needs to be assigned), so the resulting expression of Eq. 13 still holds:

$$\tilde{\boldsymbol{q}}_\tau = \mathbb{E}_\Theta(\bar{L}^\Theta_\tau \boldsymbol{e}_\Theta). \tag{17}$$





For the push-away, push-back, and push-cross pseudo-gradient vectors, we can take Eqs. 8–10 and apply the expectation operator as demonstrated in Eqs. 14–16:

$$\check{\boldsymbol{q}}_\tau = \mathbb{E}_\Theta(\check{\boldsymbol{q}}_\tau^\Theta) = \mathbb{E}_\Theta\Big( \sum_{t\in\mathcal{T}_{\tau\leftarrow}^\Theta} \bar{L}_{\tau\leftarrow t}^\Theta \boldsymbol{e}_{t\to\tau} \Big) = \sum_{t\in\mathcal{T}} \mathbb{E}_\Theta(\bar{L}_{\tau\leftarrow t}^\Theta)\boldsymbol{e}_{t\to\tau} = \sum_{t\in\mathcal{T}} \bar{L}_{\tau\leftarrow t}\boldsymbol{e}_{t\to\tau} = \sum_{t\in\mathcal{T}} \check{\boldsymbol{q}}_{\tau\leftarrow t},\tag{18}$$

$$\hat{\boldsymbol{q}}_t = \mathbb{E}_\Theta(\hat{\boldsymbol{q}}_\tau^\Theta) = \mathbb{E}_\Theta\Big( \sum_{\tau\in\mathcal{T}_{t\to}^\Theta} \bar{L}_{\tau\leftarrow t}^\Theta \boldsymbol{e}_{\tau\to t} \Big) = \sum_{\tau\in\mathcal{T}} \mathbb{E}_\Theta(\bar{L}_{\tau\leftarrow t}^\Theta)\boldsymbol{e}_{\tau\to t} = \sum_{\tau\in\mathcal{T}} \bar{L}_{\tau\leftarrow t}\boldsymbol{e}_{\tau\to t} = \sum_{t\in\mathcal{T}} \hat{\boldsymbol{q}}_{\tau\leftarrow t},\tag{19}$$

$$\dot{\boldsymbol{q}}_\tau = \mathbb{E}_\Theta(\dot{\boldsymbol{q}}_\tau^\Theta) = \mathbb{E}_\Theta\Big( \sum_{t\in\mathcal{T}_{\tau\leftarrow}^\Theta} \bar{L}_{\tau\leftarrow t}^\Theta \frac{y_{t\to\tau}^\Theta}{\|\boldsymbol{p}_{t\to\tau}\|}\boldsymbol{e}_{\Theta+\frac{\pi}{2}} \Big) = \sum_{t\in\mathcal{T}} \mathbb{E}_\Theta\Big( \bar{L}_{\tau\leftarrow t}^\Theta \frac{y_{t\to\tau}^\Theta}{\|\boldsymbol{p}_{t\to\tau}\|}\boldsymbol{e}_{\Theta+\frac{\pi}{2}} \Big) = \sum_{t\in\mathcal{T}} \dot{\boldsymbol{q}}_{\tau\leftarrow t}.\tag{20}$$

Here, the expressions after the second equality symbol correspond to applying summation over wake interactions first and expectation second. The expressions after the third equality symbol correspond to applying the expectation before the summation. This allows making the connection with Eqs. 14–16. The freedom to choose an expression may be exploited for computational reasons: depending on the wake model details either may allow for the more efficient implementation.

## 4 Optimization using pseudo-gradients

### 4.1 Overview and introduction

This section discusses how pseudo-gradients can be used for wind farm layout optimization. Here a general introduction of this topic follows. Sec. 4.2 describes proof-of-concept optimization algorithms that were used to demonstrate the viability of the approach. Sec. 4.3 shows results of the application of these algorithms to a number of academic and realistic cases. Finally, Sec. 4.4 discusses these results in general terms, disentangling the strong and weak points of the use of pseudo-gradients from the particulars of the proof-of-concept algorithms.

Sec. 3.2 already disclosed the central idea underlying pseudo-gradient based layout optimization: moving a turbine according to a pseudo-gradient attached to it will reduce the wake effect it experiences, thus resulting in a reduced wake power loss for the turbine. For simple, push-away, and push back pseudo-gradients this is because the distance between the waked and waking turbine are increased. For push-cross pseudo-gradients this is because the waked turbine is moved away from the wake centerline so as to reduce the wake incidence on its rotor plane.

For the two-turbine single wind direction case of Sec. 3.2 this is an almost trivial observation. When considering all wind directions and multiple waking or waked turbines, the resulting summed and averaged pseudo-gradient vectors as derived in Sec. 3.5 express a trade-off between the possible wind cases and wake interactions. For a given turbine, the magnitude of the resulting vector (e.g., $\|\check{\boldsymbol{q}}_\tau\|$) relative to the summed average of the magnitude of individual vectors ($\sum_{t\in\mathcal{T}} \mathbb{E}_\Theta(\|\check{\boldsymbol{q}}_{\tau\leftarrow t}^\Theta\|)$) expresses the degree of consensus on direction, including sense. For a turbine at the end of a row of turbines along the dominant wind direction, this consensus will be high, but for one in the middle of a farm at a site without a clear dominant wind direction, it will be low. For pseudo-gradient based layout optimization, the assumption is made that in any case, these resulting vectors still point in the right general direction for reducing the wake effects. Effectively, it is assumed that they can function as gradient vectors in a gradient-descent type optimization approach. This is also the reason for calling them pseudo-gradients.





So the hypothesis is that pseudo-gradient vectors can be used, after transformation to spatial vectors, to iteratively move the turbines from an initial layout to layouts of decreased (normalized) expected farm wake loss. To test this hypothesis, proof-of-concept optimization algorithms (see Sec. 4.2) were created and implemented. The hypothesis was tested for a number of cases (see Sec. 4.3). What are the advantages of using pseudo-gradients as compared to real, analytical or numerical gradients?

5        – No analytical gradients are needed. These might not be available, difficult to derive, or have to be approximated.

         – For every layout, only a single farm wake model calculation is required to produce the quantities necessary for pseudo-gradients as intermediate values, reducing the computational burden. Numerical gradients require multiple calculations of the objective to determine finite differences.

These advantages become more pronounced when more partial derivatives are involved.

Pseudo-gradients find a natural application in gradient-descent-type approaches to layout optimization, but they can be used in other approaches as well. Because of the limited computational impact of calculating them, they can be used to replace (some of) the random turbine displacement steps used in the many heuristic layout optimization approaches (e.g., Mosetti et al., 1994; Grady et al., 2005; Pookpunt and Ongsakul, 2013; Feng and Shen, 2015b; Pillai et al., 2018). This should improve the convergence speed to local optima, while the remaining random-search aspects of these approaches can preserve their
exploratory power. Even though we think that broader design space exploration can play a beneficial role in layout optimization, we do not investigate this further in this paper, to keep the focus on the strengths and weaknesses of pseudo-gradients.

## 4.2    Proof-of-concept optimization algorithms

This subsection describes three layout optimization algorithms using pseudo-gradients (Algs. 1, 5, and 7). Each one is more complex than the preceding one. The first one is the most straightforward implementation; it functions mainly as a stepping
stone to in the explanation of the other two. The second aims to improve convergence. The third furthermore aims to improve design space exploration.

Some auxiliary algorithms (Algs. 2, 3, 4, and 6) are used. They cover parts that are common to or repeated in these optimization algorithms. They are described together with the optimization algorithm they first appear in.

All algorithms start from some inputs. Among these is a valid initial layout. Approaches to creating or generating such
initial layouts are not discussed in this paper, as there is no indication that the proof-of-concept optimization algorithms depend qualitatively differently on this initial condition as compared to other optimization algorithms.

Handling of site and turbine distance constraints also forms an important part of the wind farm layout optimization problem. Again we do not discuss concrete approaches for this aspect because the specifics of constraint handling have no effect that depends on the use of pseudo-gradients. The following summary suffices: Whenever a turbine is placed outside of the site, it is
moved to the closest point on the border. Whenever two turbines become located too close to each other, they are moved away sufficiently in opposite directions. So fixing layout constraints changes the layout and affects the loss, usually increasing it.

Algorithm 1, shown below, describes an iterative optimization algorithm with a predetermined maximum number of iterations. Every iteration, first (on line 3) it calculates pseudo-gradients of predetermined type and gathers them into a so-called





layout step (making use of Alg. 2). Then (on line 4) it scales this layout step with a chosen step size and combines it with the layout to generate an updated layout (making use of Alg. 3). Finally (on line 5), it checks whether the current layout is the best one or whether it needs to terminate the optimization run early (making use of Alg. 4).

---

**Algorithm 1** Produce an optimized layout given an initial layout and pseudo-gradient type

**Input:** maximum number of iterations $n$, valid initial layout $\mathcal{L}_0$ with loss $L_0$, pseudo-gradient type $\acute{q}$, step size multiplier $s$

1: $k := 0$ {$k$ corresponds to the best layout yet encountered}
2: **for** $i := 1$ **to** $n$ **do**
3:      Use Alg. 2 to generate layout step $\acute{\mathcal{Q}}$ from $\mathcal{L}_{i-1}$ and $\acute{q}$
4:      Use Alg. 3 to update to layout $\mathcal{L}_i$ with loss $L_i$ from $\mathcal{L}_{i-1}$ and $s\acute{\mathcal{Q}}$
5:      Insert lines from Alg. 4 to update $k$ or terminate iteration early as needed
6: **end for**

**Output:** optimized layout $\mathcal{L}_k$

---

Auxiliary Algorithm 2, shown below, generates a layout step for a given layout and chosen pseudo-gradient type. It starts (on line 1) by calculating the pseudo-gradients. Then (on line 2) it removes any common shift from these pseudo-gradients, as that makes the layout drift without changing relative turbine positions. Furthermore (on line 3), it normalizes the pseudo-gradients so that the largest has magnitude one. Finally (on line 4), it gathers them in the layout step.

---

**Algorithm 2** Generate a layout step from a given layout and pseudo-gradient type

**Input:** valid layout $\mathcal{L}$, pseudo-gradient type $\acute{q}$

1: Calculate pseudo-gradients $\acute{q}_t$ for $\mathcal{L}$
2: Remove shift from pseudo-gradients: $\acute{q}'_t := q_t - \frac{1}{|\mathcal{T}|}\sum_{t\in\mathcal{T}} \acute{q}_t$
3: Normalize pseudo-gradients: $\acute{q}''_t := \acute{q}'_t / \max_{t\in\mathcal{T}} \acute{q}'_t$
4: Gather $\acute{q}''_t$ into layout step $\acute{\mathcal{Q}}$

**Output:** layout step $\acute{\mathcal{Q}}$

---

Auxiliary Algorithm 3, shown below, updates a given layout with a layout step. First (on line 1) it adds the layout step to the layout to create a new layout. Then (on line 2) it fixes any constraint violations present in this new layout. Finally (on line 3) it calculates the loss of the updated layout.

---

**Algorithm 3** Update layout and loss from a given layout and layout step

**Input:** layout $\mathcal{L}_{\text{in}}$, layout step $\mathcal{Q}$

1: Change layout: $\mathcal{L}_{\text{out}} := \mathcal{L}_{\text{in}} + \mathcal{Q}$
2: Fix any constraint violations of $\mathcal{L}_{\text{out}}$
3: Calculate loss $L_{\text{out}}$ for $\mathcal{L}_{\text{out}}$

**Output:** layout $\mathcal{L}_{\text{out}}$ and loss $L_{\text{out}}$

---





Auxiliary Algorithm 4, shown below, contains code lines to check and update the current best layout index and to decide whether the optimization run needs to be terminated early, i.e., before the maximum number of iterations has been reached. First, if the current layout's loss is smaller than the previously best layout's (line 1), the best layout index is updated (on line 2). Second, if the current layout's loss is significantly worse than the best layout's (line 3), the algorithm is terminated early (on line 4). A loss is considered significantly worse if it exceeds the best layout's loss by a fraction inversely proportional to the iteration number.

---

**Algorithm 4** Code lines to update current best layout and check early termination conditions

---

**Input:** current iteration index $i$, losses $L_k$ and $L_i$, layouts $\mathcal{L}_i$ and $\mathcal{L}_{i-1}$

1: **if** $L_i < L_k$ **then**

2:     $k := i$

3: **else if** $L_i > L_k + (L_0 - L_k)/i$ **then**

4:     **break**

5: **end if**

**Output:** best layout index $k$

---

Algorithm 5, shown below, modifies Alg. 1 by adding an adaptive step size. The aim is increasing the speed of convergence. For this, the algorithm introduces two scaling factors, which determine a small and a large step size at each iteration. Essentially (on line 4), it applies Alg. 1 for both of these step sizes. However (on line 7), it retains only the best of both resulting layouts (making use of Alg. 6). Furthermore (on line 8), the step size multiplier for the next iteration is taken to be the scaled multiplier resulting in the best layout of this iteration.

---

**Algorithm 5** Produce an optimized layout given an initial layout and pseudo-gradient type using an adaptive step size

---

**Input:** maximum number of iterations $n$, valid initial layout $\mathcal{L}_0$ with loss $L_0$, pseudo-gradient type $\acute{q}$, initial step size multiplier $s_1$, scaling factors $\alpha^-$ and $\alpha^+$

1: $k := 0$ {$k$ corresponds to the best layout yet encountered}

2: **for** $i := 1$ **to** $n$ **do**

3:     Use Alg. 2 to generate layout step $\acute{\mathcal{Q}}$ from $\mathcal{L}_{i-1}$ and $\acute{q}$

4:     **for all** $\odot \in \{-, +\}$ **do**

5:         Use Alg. 3 to update to layout $\mathcal{L}_i^\odot$ with loss $L_i^\odot$ from $\mathcal{L}_{i-1}$ and $\alpha^\odot s_i \acute{\mathcal{Q}}$

6:     **end for**

7:     Use Alg 6 to pick $\mathcal{L}_i$ with loss $L_i$ and index $\boxdot$ from $\{(\mathcal{L}_i^\odot, L_i^\odot) : \odot \in \{-, +\}\}$

8:     Rescale step size multiplier: $s_{i+1} = \alpha^\boxdot s_i$

9:     Insert lines from Alg. 4 to update $k$ or terminate iteration early as needed

10: **end for**

**Output:** optimized layout $\mathcal{L}_k$

---



Auxiliary Algorithm 6, shown below, picks the best layout from a set of layouts (with pre-computed losses). Naturally (on line 1), it selects the layout with the smallest loss.

---

**Algorithm 6** Pick best layout from set of layouts and losses

---

**Input:** set of layout–loss pairs $\{(\mathcal{L}_j, L_j) : j \in J\}$

1: $* := \arg\min_{j \in J} L_j$

**Output:** best layout $\mathcal{L}_*$ with loss $L_*$ and index $*$

---

Finally, Algorithm 7, shown below, expands on Alg. 5 by considering a set of pseudo-gradient types instead of just one. The aim is increasing its capacity to explore the space of layouts. Essentially (on line 4), it applies Alg. 5 for all the pseudo-gradient

5    types considered. But again (on line 12) it retains only the best of the resulting layouts (making use of Alg. 6). A computational analysis of this algorithm is available in App. B2.

---

**Algorithm 7** Produce an optimized layout given an initial layout using an adaptive step size

---

**Input:** maximum number of iterations $n$, valid initial layout $\mathcal{L}_0$ with loss $L_0$, initial step size multiplier $s_1$, scaling factors $\alpha^-$ and $\alpha^+$

1:   $k := 0$ {$k$ corresponds to the best layout yet encountered}

2:   $\acute{s}_1 := s_1$ for all $\acute{\boldsymbol{q}} \in \{\check{\boldsymbol{q}}, \hat{\boldsymbol{q}}, \dot{\boldsymbol{q}}\}$

3:   **for** $i := 1$ **to** $n$ **do**

4:      **for all** $\acute{\boldsymbol{q}} \in \{\check{\boldsymbol{q}}, \hat{\boldsymbol{q}}, \dot{\boldsymbol{q}}\}$ **do**

5:        Use Alg. 2 to generate layout step $\acute{\mathcal{Q}}$ from $\mathcal{L}_{i-1}$ and $\acute{\boldsymbol{q}}$

6:        **for all** $\odot \in \{-, +\}$ **do**

7:          Use Alg. 3 to update to layout $\acute{\mathcal{L}}_i^{\odot}$ with loss $\acute{L}_i^{\odot}$ from $\mathcal{L}_{i-1}$ and $\alpha^{\odot} \acute{s}_i \acute{\mathcal{Q}}$

8:        **end for**

9:        Use Alg 6 to pick $\acute{\mathcal{L}}_i$ with loss $\acute{L}_i$ and index $\boxdot$ from $\{(\acute{\mathcal{L}}_i^{\odot}, \acute{L}_i^{\odot}) : \odot \in \{-, +\}\}$

10:       Rescale step size multiplier: $\acute{s}_{i+1} = \alpha^{\boxdot} \acute{s}_i$

11:      **end for**

12:     Use Alg 6 to pick $\mathcal{L}_i$ with loss $L_i$ and index $\grave{\boldsymbol{q}}$ from $\{(\acute{\mathcal{L}}_i, \acute{L}_i) : \acute{\boldsymbol{q}} \in \{\check{\boldsymbol{q}}, \hat{\boldsymbol{q}}, \dot{\boldsymbol{q}}\}\}$

13:     Insert lines from Alg. 4 to update $k$ or terminate iteration early as needed

14: **end for**

**Output:** optimized layout $\mathcal{L}_k$

---

## 4.3   Results

### 4.3.1   Overview

This subsection shows results of the application of these algorithms to a number of academic and realistic cases. The order is

10   more-or-less from less to more complex. For all cases, a brief description of the wind resource, site, turbine and wake model





are given. All information and the scripts used to generate the results and figures are included in the code bundle made available
as supplementary material (Quaeghebeur, 2020).

The first case, in Sec. 4.3.2, is the one of the IEA Wind Task 37 wind farm layout optimization Case Study 1 (Baker et al.,
2019a). It is built up elaborately, to provide a good basis for understanding the algorithms described in Sec. 4.2. It has a simple
5  site and its wind rose is described by few directions. The second case, in Sec. 4.3.3, is one from the seminal paper of Mosetti
et al. (1994). Its wind rose has a larger number of directions. Next, in Sec. 4.3.4, comes a case built around the Horns Rev
offshore wind farm. This is the first one with a realistic wind rose. Then, in Sec. 4.3.5, the IEA Wind Task 37's reference
offshore wind farm is considered. Its non-convex site shape is more complex than those of the previous cases. Finally, in
Sec. 4.3.6, a case built around the Borssele IV site is presented. Its non-connected site is the first to be realistically complex.
For all cases, the optimization runs are set up in such a way as to get useful information about the optimization approach not
yet gleaned from the previous cases.

### 4.3.2  IEA Wind Task 37 wind farm layout optimization Case Study 1

The IEA Wind Task 37 on Systems Engineering organizes case studies to compare different approaches to wind farm layout
optimization. Baker et al. (2019a) reports on the results of Case Studies 1 and 2. Layouts produced by early versions of the
pseudo-gradient-based algorithms were submitted to this case study (Baker et al., 2019a, submissions 3 and 9). The paper
shows that the pseudo-gradient-based algorithms used are competitive in situations where computational cost is a factor (see
Baker et al., 2019a, Table 4 and Figure 5). This is the case, for example, when exploring many different starting layouts or
re-optimizing manually changed layouts.

This subsection focuses on Case Study 1, which compares algorithms for three sites, a given wind turbine, a given wind
resource, and a given wake model (cf. Bastankhah and Porté-Agel, 2014, but simplified). The sites are all disc-shaped, but vary
in size and number of turbines (16, 36, and 64). (Case Study 2 explores the effect of using different wake models, which is less
relevant here.) First the 16-turbine site is used to illustrate pseudo-gradient-based optimization and then all sites are used for
comparative purposes.

Before addressing the case with the actual wind rose used for the IEA Wind Task 37 Case Study 1, pseudo-gradient vectors
for a single wind direction are illustrated. Figure 9 shows a single-direction wind rose (on the left) and the pseudo-gradient
vectors associated to it (black vectors attached to the blue dots representing turbines), for the optional initial layout provided
as part of the IEA Wind Task 37 Case Study 1. (The site boundary is drawn in gray.) In this case study, only a single wind
speed (at rated) is considered. The simple pseudo-gradient vectors are aligned with this wind direction, by definition (cf. Eq. 2).
The push-away and push-back pseudo-gradients are constructed as a sum of inter-turbine vectors (cf. Figs. 2 and 4) that turns
out mostly, but not completely aligned with the wind direction. The push-cross pseudo-gradient vectors are orthogonal to the
single wind direction by definition (cf. Eq. 9). Below Eq. 9, it was stated that push-cross pseudo-gradient vectors have a smaller
magnitude than those of the other types. This is not visible here, because the push-cross pseudo-gradient vectors are scaled up
by a factor of 32 relative to the others.





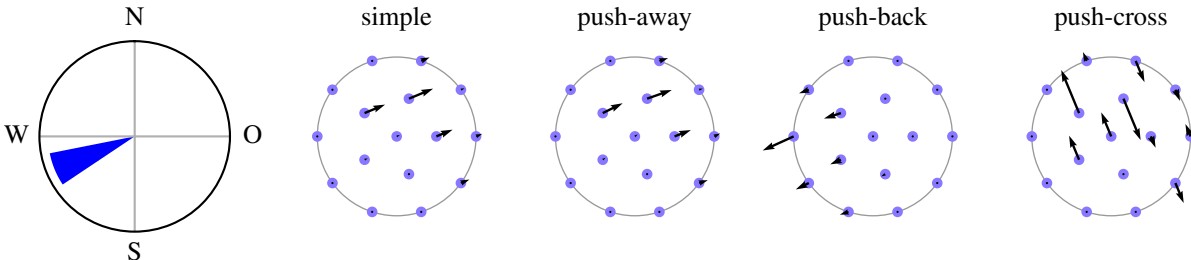

**Figure 9.** Pseudo-gradient vectors associated with a single wind direction for the IEA Wind Task 37 Case Study 1 initial layout.

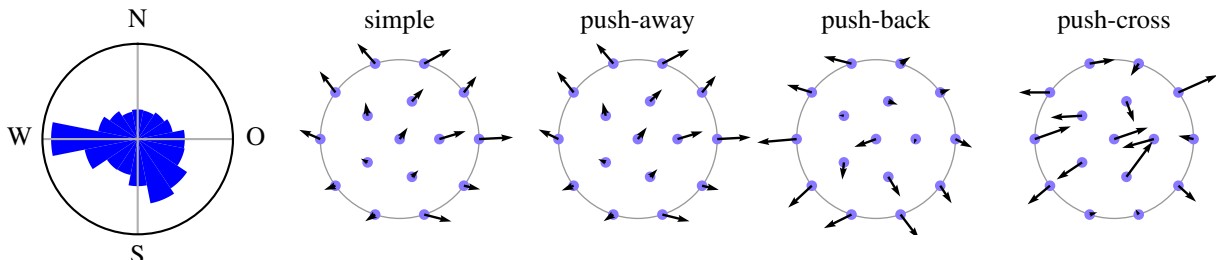

**Figure 10.** Pseudo-gradient vectors for the IEA Wind Task 37 Case Study 1 initial layout and wind rose.

Figure 10 shows the IEA Wind Task 37 Case Study 1's wind rose (on the left) and the pseudo-gradient vectors associated to it. (The wind rose wedge *area* is proportional to the direction's probability, which is less perceptually misleading than length.) The simple, push-away, and push-back pseudo-gradient vectors mostly point towards the exterior of the site. This expansionist behavior is a general tendency for these types of pseudo-gradients. The push-cross pseudo-gradient vectors do not exhibit this

behavior as much. At the end of Sec. 3.4 it was stated that the pseudo-gradient vector magnitudes after wind resource-averaging is in general reduced relative to the single wind direction-case. This is not visible here, because the vectors here are scaled up by a factor of 4 relative to the ones in Fig. 9. The vectors shown here have magnitudes larger than the spatial vectors typically used to move turbines during optimization. The maximum magnitude or step size $s$ the algorithms are initialized with typically lie between 0.5 and 3 rotor diameters $D$ (the gray dots have a diameter of $D$).

Figure 11 gives an overview of a set of 20-iteration optimization runs that where performed starting from the IEA Wind Task 37 Case Study 1 initial layout (blue dots in the top row drawings indicate initial turbine positions). The first four columns correspond to the application of Alg. 5 for each of the pseudo-gradient types listed at the top. The last column corresponds to the application of Alg. 7. The top row drawings show the evolution of the layout (black curves) and the best layout obtained (red dots circled with a gray line indicating the distance constraint). The middle row plots show the evolution of the wake loss

over the iterations (blue line) and wake loss stopping criterion value (gray, dashed line; cf. Alg. 4 line 3). The bottom row plots

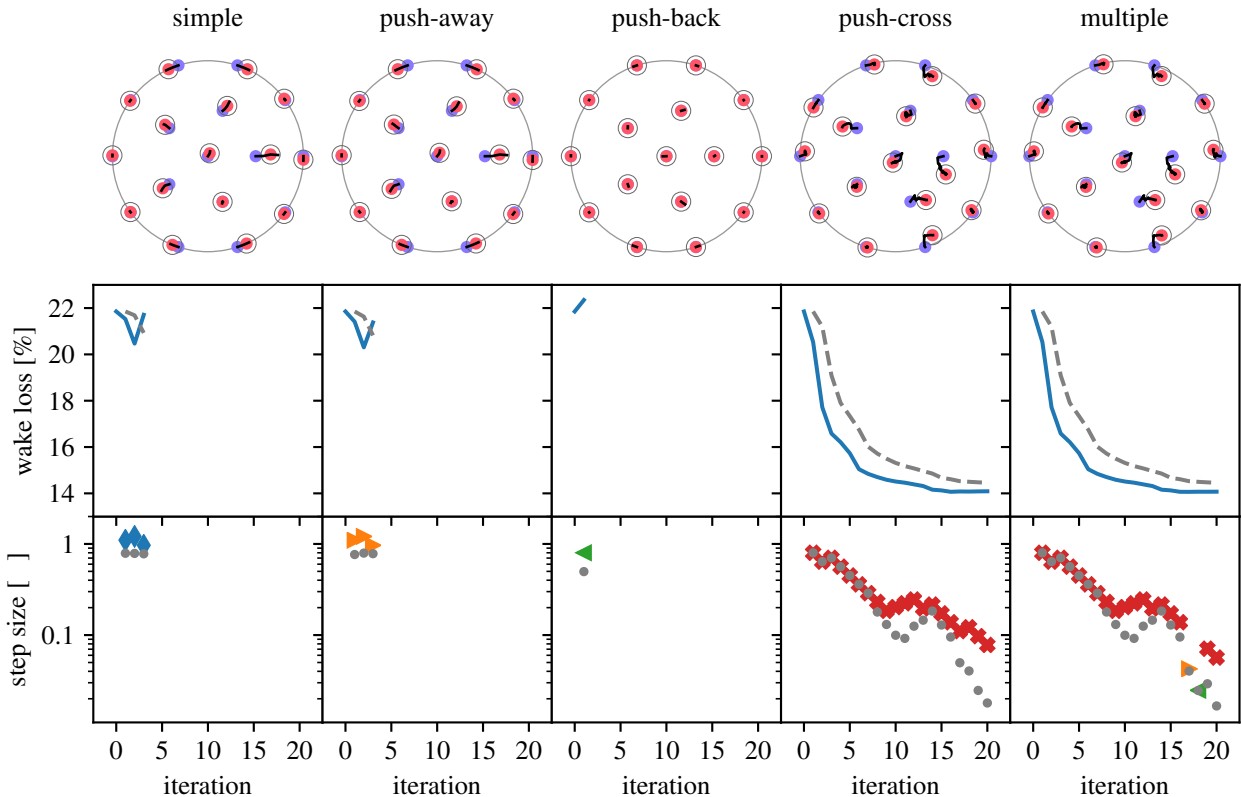

**Figure 11.** Overview of optimization runs for the IEA Wind Task 37 Case Study 1 16-turbine site using the different types of pseudo-gradients. (Legend in Table 1.)

show the maximum step size at each iteration (pseudo-gradient type-specific markers; gray dots for maximum step size after site constraint correction). The meaning of the plot elements is gathered in Table 1 for convenient reference.

The results of the optimization runs of Figure 11 vary significantly over the different pseudo-gradient types. Figures 9 and 10 showed that simple and push-away pseudo-gradients are very similar. This is reflected in the optimization runs for

5   these types, which are also very similar, although push-away pseudo-gradients perform slightly better, which is typical in our experience. The wake loss plots show that after two optimizing iterations the pseudo-gradients start having a degrading effect. The push-back pseudo-gradients are even counterproductive right at the first iteration. (They might still point in a direction of improvement, but the step size may be too large.) This is a good reminder of the fact that pseudo-gradient-based optimization, as a heuristic, provides no guarantees. However, the push-cross optimization run shows that they can be very effective indeed.

10  When using multiple pseudo-gradients and each iteration picking the best result, it is therefore no surprise push-cross pseudo-gradients dominate in this case.





**Table 1.** Legend for optimization run overview plots.

### Layout evolution drawing

- blue/red dots •/•: initial/final turbine locations
  (dot radius equal to rotor radius)

- black lines **–**: turbine path over optimization run

- gray lines —/–: site/rotor constraints

### Wake loss plot

- full blue line —: wake loss curve

- dashed gray line – –: early termination constraint

### Step size plot

- colored markers: maximal step size set

  - blue lozenge ♦: simple

  - orange triangle-right ▶: push-away

  - green triangle-left ◀: push-back

  - red cross ✕: push-cross

- gray dots •: actual maximal step size
  (after constraint handling)

For all optimization runs default parameters were used ($s_1 = D$, $(\alpha^-, \alpha^+) = (0.8, 1.1)$; cf. Alg. 5). Optimization can be improved, sometimes quite significantly, by tweaking these. Strategies for this have at this point not yet moved beyond trial-and-error.

Figure 12 gives an overview of 30-iteration optimization runs using Alg. 7 for the 36 and 64-turbines sites that where performed starting from the IEA Wind Task 37 Case Study 1 initial layouts. This time, the parameters were tweaked to both improve the optimization result and get two qualitatively different optimization behaviors. For the 36-turbine site ($s_1 = 1.3D$, $(\alpha^-, \alpha^+) = (0.5, 0.99)$) convergence is smooth and the optimized layout lies very close to the initial one. After iteration 22, the step size has become so small (points fall outside the plot) that the layout does not really change anymore: a nearby local optimum has been reached. For the 64-turbine site ($s_1 = 2D$, $(\alpha^-, \alpha^+) = (0.9, 2)$) expansionist behavior and larger steps are present, giving a non-smooth convergence. The larger steps cause exploration of a different area of the solution space, so that the final layout does not lie as close to the initial layout as for the 36-turbine site.

The outstanding success of push-cross-based optimization visible in Figs. 11 and 12 is due to the limited number of wind directions used in the wake loss calculations (16), as prescribed by the Case Study. This rough discretization of wind directions

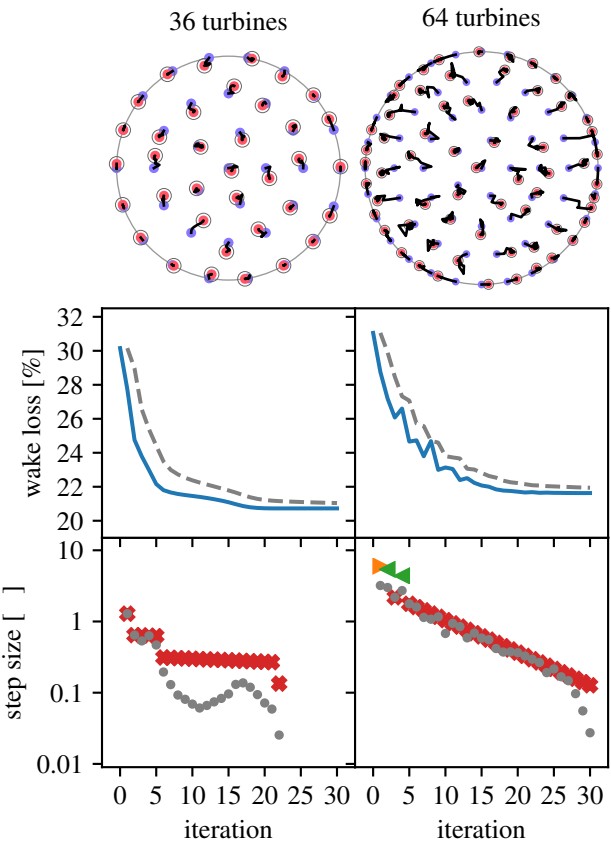

**Figure 12.** Overview of optimization runs for the IEA Wind Task 37 Case Study 1 36 and 64-turbine sites. (Legend in Table 1.)

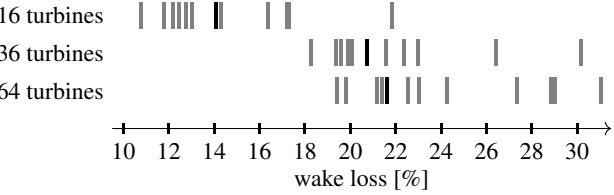

**Figure 13.** Comparison of IEA Wind Task 37 Case Study 1 wake loss percentages. Black bars show the results of the case studies discussed here.

results in 'holes' where turbines can 'hide'. These holes are artificial and do not correspond to what happens in reality (see, e.g., Feng and Shen, 2015a, Sec. 5.2). Other cases, below, do not have this defect.

Figure 13 gives a comparison of the wake loss percentages achieved by the participants in IEA Wind Task 37 Case Study 1 (gray indicators). The wake loss percentage for the layouts presented above have been added to this picture (black indicator).

5 The relative position shows decent performance of the pseudo-gradient-based optimization algorithm used, certainly consider-





ing its relative computational efficiency. Strategies compatible with pseudo-gradient-based optimization, such as using different initial layouts and applying wake spreading (Thomas and Ning, 2018) can further improve the results.

### 4.3.3 Mosetti et al.'s problem

Mosetti et al. (1994) wrote a seminal paper on wind farm layout optimization. They used a genetic algorithm for a discretized

solution space. The problems they analyze have been used as a benchmark by many others (e.g., Grady et al., 2005; Pookpunt and Ongsakul, 2013; Turner et al., 2014; Feng and Shen, 2015b; Pillai et al., 2018). Here, their multiple wind direction and multiple wind speed problem (Mosetti et al., 1994, Sec. 4.3) is considered, using Jensen's model as described by Frandsen (1992, Sec. 2.2) with rotor-plane averaging, 36 wind directions, fifteen 630 kW turbines, and a square site. In the follow-up literature cited, the wind rose used was inadvertently modified. We use the original wind rose, so only a comparison with the

results of the original paper is made.

    Figure 14 gives an overview of the optimization runs performed for the selected problem. The (area-proportional) wind rose is at the top. The left column shows the result of an optimization run starting from Mosetti et al.'s optimized layout ($s_1 = 3D$, $(\alpha^-, \alpha^+) = (0.9, 1.1)$). The higher number of wind directions as compared with the problem of Sec. 4.3.2 makes push-cross pseudo-gradients much less attractive, leading to only push-away and push-back pseudo-gradients being used. The resulting

expansionist behavior leads to an optimized layout with all but one turbine at the site's border. Because of the low power density, the wake loss is quite small for this problem. Nevertheless, a significant relative improvement can be made. The right column shows the result when starting from a regular hexagonal layout ($s_1 = 3D$, $(\alpha^-, \alpha^+) = (0.7, 1.3)$). It proves that it is possible to achieve similar results when starting from a non-optimized initial layout. The optimization evolution and optimized layout are symmetric relative to the dominant wind direction due to the symmetry of the wind rose around this direction and

the (almost) alignment of the hexagonal layout with this direction. (When fully aligning the hexagonal layout, the resulting initial layout had a low wake loss of about 2.7 %. Optimization then failed, likely because this initial layout corresponds to a deep local minimum.)

### 4.3.4 Horns Rev 1

Horns Rev 1 is well-known, as the first large-scale offshore wind farm. The site has the shape of a parallelogram. The farm

is composed of 80 V80-2.0 MW turbines. Here the wind farm layout optimization problem for this site as defined by Feng and Shen (2015b, Sec. 5, Case 1) is considered, using Jensen's model with rotor-plane averaging. It subdivides a 12-direction wind rose into 360 wind directions (using nearest-neighbor interpolation), which makes it far more realistic than the problems discussed above. It uses a minimal inter-turbine distance $d_{\mathrm{mit}}$ of $5D$, which implies that the turbines have less room to maneuver than in the problems discussed above.

Figure 15 gives an overview of the optimization runs performed for the selected problem. The (area-proportional) wind rose is at the top. The left column shows the result of an optimization run starting from the actual Horns Rev 1 layout ($s_1 = 0.5D$, $(\alpha^-, \alpha^+) = (0.7, 1.1)$). As for the problem of Sec. 4.3.3, there is clear expansionist behavior. During the expansion phase, driven mostly by push-back pseudo-gradients, the greatest improvement is seen. However, because of the limited maneuvering



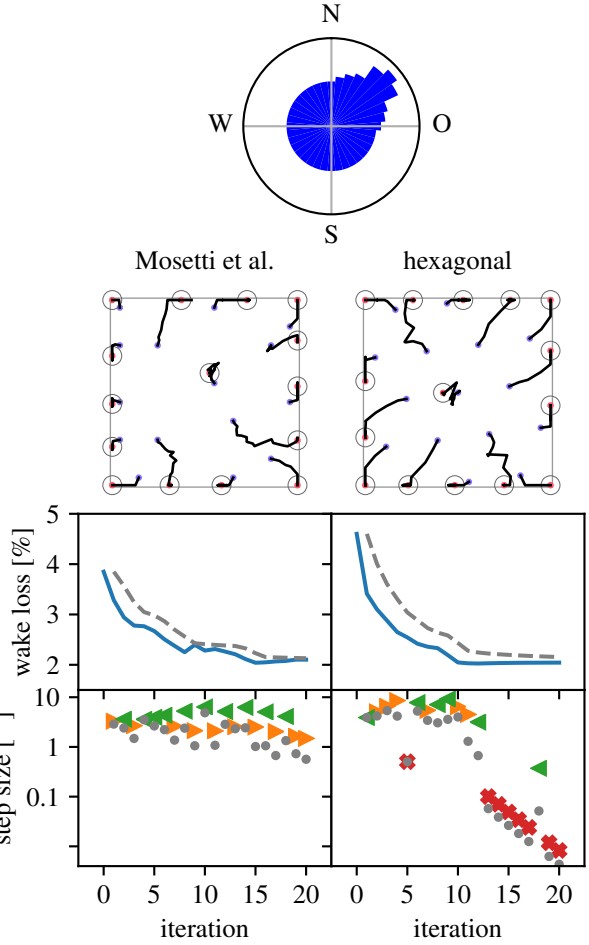

**Figure 14.** Overview of optimization runs for the Mosetti et al. (1994) wind farm layout problem. (Legend in Table 1.)

space, further improvement attempts mostly involve push-cross pseudo-gradients. Given the inter-turbine constraint, the original Horns Rev 1 layout appears well-optimized already, because little improvement can be made. The right column shows the result when starting from a regular hexagonal layout ($s_1 = 2D$, $(\alpha^-, \alpha^+) = (0.8, 1.1)$). It proves that it is possible to achieve similar results when starting from a non-optimized initial layout. The optimization behavior mimics that of the run for the

5  original layout, but with larger step sizes. The most important qualitative difference is a higher turbine density on the site border.

(A comparison with the results of Feng and Shen (2015b, Sec. 5, Case 1) was not possible. Namely, the wakeless power obtained for the original layout differs from theirs, and so do the wake loss values, and sufficiently so that wake loss values for optimized layouts cannot be reliably compared. While very helpful, Feng and Shen could not provide us the materials needed

10  to determine the cause of the difference.)

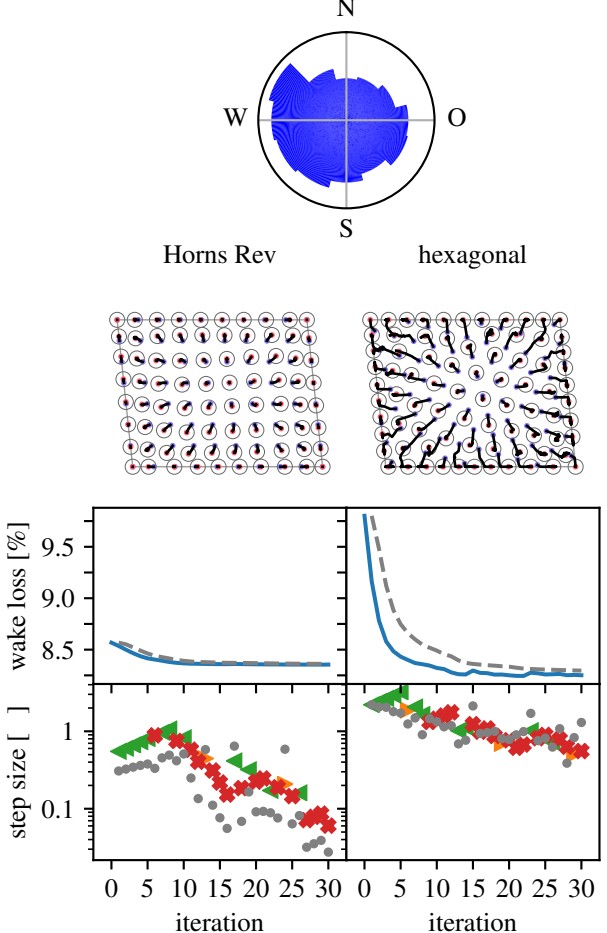

**Figure 15.** Overview of optimization runs for the Horns Rev wind farm layout problem. (Legend in Table 1.)

### 4.3.5 IEA Wind Task 37 reference offshore plant

The IEA Wind Task 37 on Systems Engineering is defining a reference offshore plant. This is a description of an offshore wind farm meant to serve for comparisons of offshore wind farm design tools, i.e., for benchmarking. It goes beyond simple power-based layout optimization, as covered in this paper, and considers cable layout and substructure costs as well. Sanchez

5 Perez-Moreno (2018) provides the actual definition. New in this paper is that the site is non-convex. The farm is composed of 74 reference 10 MW turbines. It subdivides a 16-direction wind rose into 360 wind directions (using linear interpolation) and uses Jensen's model with rotor-plane averaging. It uses a minimal inter-turbine distance $d_{\mathrm{mit}}$ of $3D$, which makes for a much sparser layout problem than the one for Horns Rev above.



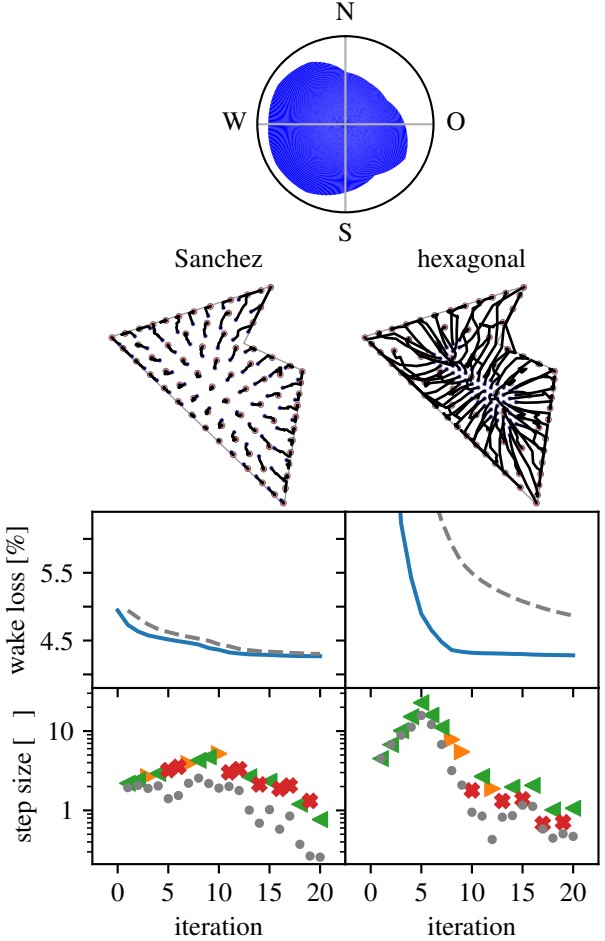

**Figure 16.** Overview of optimization runs for the IEA Wind Task 37 offshore reference wind farm layout problem. (Legend in Table 1.)

Figure 16 gives an overview of the optimization runs performed for the selected problem. The (area-proportional) wind rose is at the top. The left column shows the result of an optimization run starting from the reference layout ($s_1 = 2D$, $(\alpha^-, \alpha^+) = (0.8, 1.1)$). The right column shows the result when starting from a regular hexagonal layout ($s_1 = 3D$, $(\alpha^-, \alpha^+) = (0.7, 1.5)$) constrained to the central area of the site. For both cases, the optimization behavior is similar to the one seen in Fig. 15 and there is a significant relative improvement. The right-column result shows that the algorithm has no problem with the irregular, non-convex shape and manages to place turbines in every part of it.

### 4.3.6 Borssele IV

The final problem considered in this paper is one constructed on the basis of the Borssele IV site. The new aspect this site brings to the table is that it is composed of multiple non-connected parcels. The Dutch government has published a detailed



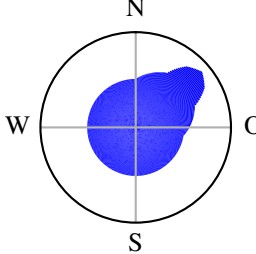

**Figure 17.** The wind rose used for the Borssele IV cases.

description of this actual site (RVO, 2016; van der Heijden, 2016). The wind resource used is Mosetti et al.'s (cf. Sec. 4.3.3), but now with 360 wind directions (obtained using linear interpolation); see Fig. 17. This case uses Jensen's model with rotor-plane averaging. The turbine used is the 10 MW IEA37 offshore reference one (cf. Sec. 4.3.5). The minimal inter-turbine distance $d_{\mathrm{mit}}$ is $4D$.

This problem is used to explore the effect of different turbine densities and scaling parameters on the optimization. Layouts with 30, 50, 70, and 90 turbines are considered. The parameters $(\alpha^-,\alpha^+)=(0.9,1.1)$ define a 'soft' scaling strategy that allows for only small differences in step size between consecutive iterations. The parameters $(\alpha^-,\alpha^+)=(0.5,2)$ define an 'aggressive' scaling strategy that forces substantial differences in step size between consecutive iterations. For all cases, $s_1 = 2D$.

Figure 18 gives an overview of the resulting optimization runs. It shows that the pseudo-gradient approach works with a wide variety of turbine densities. For increasing densities, the wake loss increases, of course, but the algorithm always manages to achieve a significant improvement. It also shows that differences in scaling strategy have a clear impact on the optimization behavior. Most notably, the 'aggressive' strategy manages to move turbines between parcels, whereas the 'soft' strategy does not (cf. 30, 70, and 90 turbine cases). This can result in a noticeable improvement.

## 15  4.4  Discussion

This section provides a discussions of the results from two perspectives. The first, academic perspective, considers the proof-of-concept algorithm and results presented in the sections above. The aim is to disentangle the strong and weak points of the use of pseudo-gradients from the particulars of the proof-of-concept algorithms. The second, industry perspective, considers the non-public counterparts of the algorithm and results. The aim is to share, in general terms, the experience gained and lessons

learned from its practical application.

## 4.5  The academic perspective

The proof-of-concept algorithms of Sec. 4.2 are all purely deterministic. Randomized steps in the design space and the use of multiple candidate solutions evolving in parallel are an important driver of the exploratory power of many heuristic optimiza-

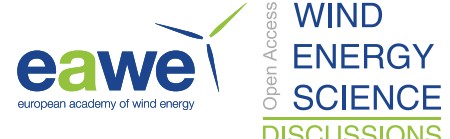

(a) 30 turbines.

(b) 50 turbines.

(c) 70 turbines.

(d) 90 turbines.

**Figure 18.** Overview of optimization runs for the Borssele IV wind farm layout problem. (Legend in Table 1.)





tion algorithms. Here, exploratory power is created by using multiple pseudo-gradients concurrently and each step picking the one that delivers the best results (cf. Alg. 7). To improve the exploratory power, the above-mentioned techniques from heuristic optimization can be added. Based on comparisons with other approaches (cf. Sec. 4.3.2 and specifically Fig. 13), this may be necessary to be able to achieve results comparable to the current best performing algorithms. This would trade off

computational speed for exploratory power.

Because of the gradient-like nature of pseudo-gradients, the proof-of-concept algorithms can also be extended to make use of innovations for gradient-based optimization methods. The already-included use of an adaptive step is one example. The technique of wake-spreading helps avoid shallow local minima (Thomas and Ning, 2018). It can be directly integrated. Because it increases the number of iterations necessary for convergence, it trades off computational speed for convergence quality.

The fact that these techniques from heuristic and gradient-based optimization theory were not applied for this paper's study is intentional. It makes the results presented (Figs. 12, 14, 15, 16, 18) show very clearly that pseudo-gradient-based optimization can achieve significant layout improvements in a very limited number of iterations. The main reason for this is the following: Contrary to most existing heuristic methods, but similar to gradient-based optimization, the steps taken each iteration are purposeful, being constructed from domain knowledge. However, contrary to gradient-based approaches, there is no need to

calculate derivatives.

That does not mean that pseudo-gradient-based optimization provides the best of both worlds. Specifically, the heuristic nature of the pseudo-gradients and their essentially decentralized computation (one per turbine) imply that they will not be able to match true gradients in their ability to point towards the objective's optima. An important unanswered question here is a quantification of this difference. On the other hand, existing heuristic approaches could all benefit from replacing some of

the random steps or part of each random step by pseudo-gradient-based steps.

Looking at the pseudo-gradient-based steps in Figs. 9 and 10 and the step types in all the overviews (Figs. 11, 12, 14, 15, 16, 18), it becomes clear that there are two qualitatively different classes of pseudo-gradients. Namely, there are the push-cross ones versus the other, outward pushing ones. The latter lead to an optimized use of the available space in the site. The former optimize the relative position of the turbines to reduce wake incidence. The expansionist behavior resulting from the

outward pushing ones is understandable, but leads to the turbines bunching up near the border, precluding proper exploration of more uniform layouts. For cases with a realistic number of wind directions, push-cross pseudo-gradients become important only after the initial stage of the optimization run, when the most substantial improvement is seen. By then, many turbines have bunched-up on or near the border, reducing their freedom of movement and therefore the possible efficacy of push-cross pseudo-gradients.

There are potential options for improving the effectiveness and usefulness of pseudo-gradients. In the proof-of-concept algorithm, the different pseudo-gradient types are applied in an either-or fashion. One might temper the expansionist behavior of the outward pushing pseudo-gradient vectors by mixing in push-cross pseudo-gradient vectors (making linear combinations). Also, strategies for scaling a turbine's step size depending on its distance to the border can be devised, for example to control expansionist behavior. (This may create a coupling to the site constraint handling.) Furthermore, push-cross pseudo-gradients





can be used for wake steering through yaw control instead of or next to turbine displacement, as that is also used for reducing wake incidence (see, e.g., Fleming et al., 2016).

The parameter settings for the proof-of-concept algorithm have a clear impact on their optimization behavior (cf. Fig. 18). This shows that flexible layout optimization algorithms can be devised based on pseudo-gradients. This is not specifically linked
to the specific nature of pseudo-gradients, as similar flexibility can be achieved using, e.g., real gradients. Many other ways of making the algorithm more flexible can be thought of. For example by adding functionality to only move one or a subset of turbines, which can allow for a reduction in the per-iteration computational complexity at a cost of slower convergence.

Despite enabling effective and efficient layout optimization algorithms, there are two important properties that pseudo-gradients cannot provide. First, they are defined locally for each turbine based on a proxy for the objective function. This
makes optimization partially blind to this objective. (The objective is of course used to select between different types of pseudo-gradients, but that is due to the design of the proof-of-concept algorithm.) Gradient-based optimization does not have this downside. Second, they require a starting layout, although this is the case for most existing layout optimization approaches. That makes the optimization depend quite strongly on the initial layout. Algorithms that construct a layout by placing one turbine at a time do not have this problem (see, e.g., Changshui et al., 2011; Tilli, 2019). Of course such algorithms can be used
to create a starting layout for pseudo-gradient-based optimization.

## 4.6   The industry perspective

In an industry environment, the algorithm was successful in creating layouts that performed as well as those created with commercial software packages, but at a fraction of the runtime. Because the wind turbines gradually move towards their optimal location over the course of the iteration steps, it also gives a design team good insight in how the optimization progresses and
whether it matches engineering intuition. This is important in order to catch errors and weaknesses in the cost function—which can lead to severely biased results—but also to be able to defend the results. Very artificial-looking layouts that are produced by a black box-type algorithm are often scrutinized and have more difficulties being accepted in a business environment.

One of the major challenges is how to deal with practical location constraints. Within the site boundaries, an offshore site usually has areas where wind turbines cannot be placed. For example, shipwrecks and war graves are often surrounded by
buffer zones where offshore activities are forbidden, and a designer may choose to avoid (clusters of) obstacles that are too cumbersome to remove. There may also be areas where the soil type makes it impracticable to install a foundation, or where sand banks limit the accessibility of large vessels. Moreover, some sites (e.g., Borssele) are crossed by existing (telecom) cables, pipelines, or shipping lanes that each have safety zones of usually 500 m. Combined, this often leads to a location constraint polygon that is concave, with multiple regions, and with numerous holes. A pseudo-gradient algorithm that moves
turbines around therefore needs to contain a rationale on when to cross certain zones or how to navigate around obstacles. A combination with a tangent bug algorithm has proven to be successful in the past, but other solutions undoubtedly exist.




## 5 Conclusions

The pseudo-gradient concept is useful for wind farm layout optimization. Pseudo-gradients can be derived efficiently during the wake loss calculations necessary to evaluate a layout (Sec. 3). It is straightforward to build a wind farm layout optimization algorithm using them (Sec. 4.2). Such algorithms have proven themselves effective, versatile, and efficient (Sec. 4.3). Because

of their computational efficiency, pseudo-gradients-based algorithms are an enabler for analyses, such as robustness studies, that require a number of iterations or repetitions that make many other approaches computationally prohibitive. They do have their weaknesses, such as their strong dependence on an initial layout and a limited exploratory power, leading, e.g., to layouts with many turbines on the border. There are also limitations, such as simple pseudo-gradients being available for computational fluid dynamics-based wake models.

The pseudo-gradient concept is flexible. Pseudo-gradients can be defined for a wide range of wake models (Sec. 2.5). It is in principle applicable also beyond wakes to other air-mediated turbine interactions, such as induction and blockage, as long as a per-turbine loss (or perhaps gain) can be obtained from the interaction model. Even other layout optimization-relevant aspects such as the impact of water depth and cable interconnections for offshore wind farms allow for a pseudo-gradient-type treatment. (This has been done in a non-public implementation of the second author.) The only things that are needed to create

such pseudo-gradients are an indicator of the magnitude of a favourable or unfavourable performance indicator, and one or more (possibly intuitive) definitions of directions in which improvements are expected. Focusing again on wake models, different pseudo-gradient variants can be defined (Sec. 3), leading to qualitatively different behavior during optimization (Sec. 4.3). While this paper presents gradient-following algorithms (Sec. 4.2), pseudo-gradients could also be used to replace random steps in typical heuristic optimization approaches (e.g., genetic and particle swarm algorithms).

There are many possible further investigations that can start from the ideas presented in this paper. The following has already been mentioned: integration in various heuristic optimization approaches, the definition of new pseudo-gradient variants, and the combination of pseudo-gradient vectors for potentially more effective optimization. Two further ideas related to research of current interest to the wind energy community are related to push-cross pseudo-gradients:

– When adding hub height as a design variable (see, e.g., Stanley et al., 2017), push-cross pseudo-gradients might be useful
for the optimization of the height of individual turbines.

– For wake steering (see, e.g., Fleming et al., 2016), push-cross pseudo-gradients may be used to tune yaw-misalignment of each turbine for each wind direction.

Finally, the public development of pseudo-gradient (compatible) approaches to aspects of the multi-disciplinary wind farm layout optimization problem (cable layout, substructure cost, etc.) is necessary for the continued relevance of the concept in
the wind energy community.

*Code and data availability.* The implementation code and data used to define all problem cases is publicly available (Quaeghebeur, 2020).



## Appendix A: Mathematical details

### A1 Expectation for the wind resource

Let the marginal probability mass function for $\Theta$ be denoted $p_\Theta$ and the conditional probability density or mass functions for $U^\Theta$ be denoted, respectively, $f_{U^\Theta}$ or $p_{U^\Theta}$. Usually, the conditional wind speed probability density functions $f_{U^\Theta}$ are

Weibull distributions and the conditional probability mass functions $p_{U^\Theta}$ can be discretizations thereof or derived directly from a wind speed dataset.

The expectation of a function $g$ that depends on wind direction $\Theta$ and possibly other variables $o$ can then be calculated using the following expression:

$$\bar{g}(o) = \mathbb{E}_\Theta\big(g(\Theta, o)\big) = \sum_{\theta \in \Omega_\Theta} g(\theta, o) p_\Theta(\theta),$$

where $\Omega_\Theta$ is the set of discrete wind directions considered. Similarly, the conditional expectation of a function $g$ that depends on wind speed $U^\theta$ for a given direction $\theta$ and possibly other variables $o$, can then be calculated using the following expression:

$$\bar{g}^\theta(o) = \mathbb{E}_{U^\theta}\big(g(U^\Theta, o)\big) = \begin{cases} \int_0^\infty g(u, o) f_{U^\theta}(u)\, \mathrm{d}u & \text{(continuous wind speed case)}, \\ \sum_{u \in \Omega_{U^\theta}} g(u, o) p_{U^\theta}(u) & \text{(discrete wind speed case)}, \end{cases}$$

where $\Omega_{U^\theta}$ is the set of discrete free stream wind speeds considered. Finally, to calculate the joint expectation of a function $g$ that depends both on wind direction $\Theta$ and wind speed $U^\Theta$, we apply the law of the iterated expectation:

$$\mathbb{E}\big(g(U^\Theta, \Theta, o)\big) = \mathbb{E}_\Theta\big(\mathbb{E}_{U^\Theta}\big(g(U^\Theta, \Theta, o)\big)\big) = \mathbb{E}_\Theta\big(\bar{g}^\Theta(\Theta, o)\big) = \bar{g}(o).$$

## Appendix B: Computational considerations

### B1 Calculating the wake wind speed

Consider once more the turbine-specific representative inflow wind speed $U_\tau^\Theta$. It is the result of a number of nontrivial calculation steps. The expansion of its defining expressions (see Sec. 2.5) provides useful insight:

$$U_\tau^\Theta = b\big(U^\Theta, \{\mathcal{U}_{\tau \leftarrow t}^\Theta : t \in \mathcal{T}_{\tau \leftarrow}^\Theta\}\big) = b\Big(U^\Theta, \{\mathcal{W}(U_t^\Theta, U^\Theta, \ell_{t \rightarrow \tau}^\Theta, \mathcal{R}) : t \in \mathcal{T}_{\tau \leftarrow}^\Theta\}\Big).$$

These expressions' dependence on the wind direction $\Theta$ is very explicit. Furthermore, the last expression shows that the representative inflow wind speed $U_t^\Theta$ at the waking turbines needs to be available.

When the turbines can be linearly ordered such that a turbine only wakes others that come later in the order, the calculation of the speeds $U_\tau^\Theta$ can be performed in that order. So then the above expression still provides an explicit calculation procedure.

Because this ordering depends on the wind direction, this is a second, implicit way in which $\Theta$ has an effect. However, to simplify the calculations, $U_t^\Theta$ is often replaced by $U^\Theta$ in the wake model $w$. (This was also done when deriving the results for this paper.) This makes the representative inflow wind speed calculations for a turbine independent from the representative inflow wind speed of others, facilitating parallel computation.




## B2 Computational analysis of the proof-of-concept algorithm

Consider Alg. 7. This section discusses the computational cost of all parts of the algorithm.

The outer loop starting on line 3 regulates the optimization iteration. There are $n$ iterations and as in all iterations (mostly) the same computations are performed, this means the computational cost is linear in $n$. Because each iteration depends on the outcome of the previous iteration, this loop cannot be parallelized.

The loop over pseudo-gradient variants starting on line 4 requires a repetition of three times essentially the same computation, so the the computational cost for the computations it contains must be multiplied by 3. Because the results of one computation do not depend on another, this loop can be fully parallelized.

The same argument holds for the loop over two step scalings starting on line 6. This means the cost for the computations it contains must be multiplied by 2, but again that this can be done fully parallelized.

The effect of the loops is now clear and we can write an expression for the computational cost as a function of the cost $c_{7;i}$ for each of Alg. 7's lines $i$: $n(c_{7;5} + 3(2c_{7;7} + c_{7;9} + c_{7;10}) + c_{7;12} + c_{7;13})$. Most of these costs correspond to the cost of running an auxiliary algorithm, so replace the indices to make this explicit: $n(c_2 + 3(2c_3 + c_6^{(2)} + c_{7;10}) + c_6^{(3)} + c_4)$. Here, Alg. 6, which picks the minimum from a finite set of values, appears twice, once for a set of two and once for a set of three values. Relative to the other costs, these are insignificant. Similarly, the cost for line 10 of Alg. 7, multiplying two values, can also be ignored. The same holds for Alg. 4, which corresponds to the comparison of two numbers. This leaves us with $n(c_2 + 6c_3)$.

So Alg. 2 and Alg. 3 need to be investigated further. The former consists of a pseudo-gradient calculation step and then a number of arithmetic operations on the pseudo-gradient vectors. The latter consists of an arithmetic operation on the layout, constraint handling, and a wake loss calculation. The arithmetic operations are all applied to arrays of $|\mathcal{T}|$ 2-component vectors and their cost is therefore proportional to $2|\mathcal{T}|$. The site constraint handling must be done for each turbine, so has a cost proportional to $|\mathcal{T}|$ as well, but now the proportionality constant depends on the complexity of the site and may be significant relative to $|\mathcal{T}|$. The safety distance constraint must be handled for every pair of turbines and therefore has a complexity proportional to $|\mathcal{T}^2|$. The wake loss calculations also involve pairs of turbines and next to that the calculation of an expectation over the wind resource (cf. App. B1), which leads to a cost essentially proportional to $|\mathcal{T}^2|$, the number of wind directions $|\Omega_\Theta|$, and the number of wind speeds $|\Omega_U|$. The pseudo-gradient calculation is similar in complexity to the wake loss one.

Combining the results of the two preceding paragraphs gives an expression for the computational cost of the form

$$n(\alpha_{\text{pseudo-gradient}}|\mathcal{T}^2 \times \Omega_\Theta \times \Omega_U| + \alpha_{\text{arithmetic}}|\mathcal{T}| + 6\beta_{\text{arithmetic}}|\mathcal{T}| + 6\beta_{\text{site}}|\mathcal{T}| + 6\beta_{\text{safety}}|\mathcal{T}^2| + 6\beta_{\text{wake}}|\mathcal{T}^2 \times \Omega_\Theta \times \Omega_U|).$$

In practice, the arithmetic operations do not play a significant role. So, grouping terms, the computational cost picture can be summarized by $n(\gamma|\mathcal{T}^2 \times \Omega_\Theta \times \Omega_U| + \gamma_{\text{site}}|\mathcal{T}| + \gamma_{\text{safety}}|\mathcal{T}^2|)$. In this expression the last two, constraint-related terms have an important impact in practice, but are outside the scope of this paper. The first term shows that the computational cost scales quadratically with the number of turbines and linearly with the number of wind directions and wind speeds. To manage the turbine-count related complexity, an option is to only move a limited number of turbines each iteration (cf. Wagner et al., 2013, Sec. 3.1). To manage the complexity related to the number of wind directions and wind speeds, a pre-averaging-type approach is an option (cf. Tilli, 2019).





*Author contributions.* Erik Quaeghebeur developed the idea of pseudo-gradients for layout optimization, implemented it, and applied it to academic problems. He wrote most of the paper. René Bos had independently thought of the concept and was prompted by a talk of Erik's at a EUROS Meeting (cf. Acknowledgements) to revisit the idea. He created his own (much extended) implementation and applied it to industry problems. He wrote the introduction and industry-perspective discussion. He also provided feedback and suggestions on the paper.
5   René and Erik furthermore discussed their particular experience and implementations, influencing each other's further development. Michiel Zaaijer provided Erik with extensive feedback and substantive suggestions throughout the development and implementation process. He also assisted with a thorough revision of the draft paper.

*Competing interests.* The authors declare that they have no competing interests.

*Acknowledgements.* This research is part of the Dutch EUROS program, which is supported by NWO domain Applied and Engineering
10   Sciences and partly funded by the Dutch Ministry of Economic Affairs.

We are grateful for Ju Feng (DTU) for providing the wind resource data for the Horns Rev layout optimization problem (cf. Sec. 4.3.4), answering questions about their paper (Feng and Shen, 2015b), and for their efforts digging up other materials.





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
