# Peer review of "Wind farm layout optimization using pseudo-gradients"

_Wind Energy Science, 2020_

## Referee Comment (RC1) · Anonymous Referee #1 · 29 Nov 2020

After a first look to the paper, I accepted to contribute with my comments since I considered the issue highly relevant for the wind energy community and because I though, after that preliminary look, that I could contribute with useful comments about the wake models used, the implemented wind turbine models or the goal optimisation functions selected. After a first look to the paper I accepted to contribute with my comments since the issue is highly relevant for the wind energy community and because I though, after that preliminary look that I could contribute with useful comments about the wake models used, the implemented wind turbine models or about the goal optimisation functions selected.

After a detailed reading of the work, I have realised that the paper is 90% focused of purely abstract aspects of the new proposed optimisation method and associated

algorithm.

The paper contains very few, and also highly abstract, references to the physical models implemented (wakes, wind turbines). The authors present results on wake losses reductions after the application of their model to different well established cases studies, and they seem very promising.

However, in my opinion the paper should be mainly reviewed by experts in pure abstract aspects of optimization algorithms who would determine if this approach is really new, relevant and efficient as the authors claims. Personally I only could contribute with general comments (positive in general).

Without a detailed analysis of thet abstract mathematical details of the proposed model, I would say that the issue is relevant since, as the authors claim, their optimisation method does not require massive evaluation of goal functions, that can be computationally expensive in wind farm layout optimisation process.

They conceptually compare the proposed method with other approaches for wind farm layout optimisation, such as genetic algorithms (which require a high number of goal function evaluations). However, they do not include, for instance, a quantitative comparison about computational times or wind farm efficiency gains. This comparison would improve the quality of the paper.

The presentation of the models seems to be detailed and complete from a formal point of view, with an acceptable number of references.

On the other hand, the formal presentation quality seems high. The scientific results and conclusions presented are well written, clear and concise. Figures and tables are well presented.

My opinion on this paper would not determine a final acceptance/rejection decision. A deep review by expert's eyes of the optimisation method formalism is recommended. If this review if positive, I would recommend to accept the paper.

---

## Referee Comment (RC2) · Andrew P. J. Stanley (Referee) · 20 Jan 2021

The presented manuscript provides a computationally efficient method to perform wind farm layout optimization. The pseudo-gradients estimate the behavior of the design space to help find an improved wind farm layout using traditional gradient-based optimization methods.

I think the paper is very interesting and useful! Below are a list of questions/comments that I think would be helpful to clarify or consider.

1. In the overview of the pseudo-gradients (starting with section 3.1), it is not intuitive to me why you chose three pseudo-gradient definitions to apply to the waked turbine, and only one to the waking turbine. It seems like the three pseudo-gradients would

overdefine the downwind turbine and limit the upwind. Why not use two for each?

2. This may be related to the first question, so they may both be clarified in one step. When I first read through I understood (or maybe just assumed) that the pseudo-gradients would be combined to determine the step size and direction during optimization. However, when reading through the results it seemed like you chose the one pseudo-gradient type that achieved the best gain, and only took a step according to that pseudo-gradient. Which is correct? If you do only use one pseudo-gradient to determine the step, why not combine them to achieve better/faster convergence?

3. Not an issue, more a curiosity that other people might have as well. Have you considered including the constraint information in the pseudo-gradients as well, as to not need to force feasible solutions and potentially gain more exploration?

4. Reading through the manuscript, my impression is that the objective function really doesn't matter that much, as the pseudo-gradient direction is determined spatially and the step size is not really the true gradient. Have you tried a similar method with just spatial information? You would just need one or two tuning parameters that would be used to determine step size as a function of distance, but you wouldn't need a wake model. Has this been explored at all? (note, I don't necessarily think this needs to be fully explored or resolved for this publication, but I do think others might have a similar question so it might be worth addressing in the text)

5. The magnitude of the cross-stream pseudo-gradient seems arbitrary. Using a projection of one of the other pseudo-gradients (if I understood correctly) means the magnitude would always be relatively small. Often though the cross-stream gradients would be much higher than the stream wise ones (moving side to side gives much faster gains than moving further apart). It seems your formulation prioritizes that opposite. Can you explain or justify this?

6. It seems like a main outcome of this work is to be able to quickly and efficiently perform wind farm layout optimization that finds an improved layout, but much faster than

other methods. However, you haven't provided any of that comparison. You probably don't need it for everything, but for one or two of the cases can you provide comparisons of optimizer performance and computational expense with some other methods? It seems like from a performance perspective, optimizing with exact analytic gradients would be just as fast as your proposed method, and achieve a better result. That would certainly not discount using pseudo-gradients to perform layout optimization, as often exact analytic gradients are difficult and time consuming to derive, but it would help frame it better.

A few small notes

- I recommend changing the bullets on page 3 (lines 25-30). They look like minus signs.

- Page 7 line 6: should be "this value is the same for all turbines"?

- Page 7 line 17: clarify that they are equivalent because you assume a fixed number of turbines.

- Page 8 line 1: "away" and "back" were initially unclear to me. It would be helpful to clarify, maybe with relative terms? "Further downstream" and "further upstream" could potentially work.

---

## Author Comment (AC1) · 28 Mar 2021

March 28, 2021

This response to reviews is done per reviewer and then follows the general sectioning structure of the paper (so we may reorder some comments), but first the general comments are treated. The format is as follows:

- The reviewer comment is shown in an upright font.

  *Our response follows in blue and slanted font.*

  *The planned changes are written in red and slanted smaller font.*

**RC1 (Anonymous)**

General comments

- [. . .] in my opinion the paper should be mainly reviewed by experts in pure abstract aspects of optimization algorithms who would determine if this approach is really new, relevant and efficient as the authors claims.

  *We feel a review by an expert in the general field provides value, even if they cannot judge algorithmic details or sub-field relevance. We agree that scrutiny*

*of experts on wind farm layout optimization is desirable and this is provided by the second reviewer. We think it is up to the editor to decide whether sufficient expertise is covered by both reviews. Outside of the review process for this paper, we have participated using the proposed algorithm in IEA Wind Task 37's Case Studies on wind farm layout optimization. This is not the same as a review, but does provide a check of the algorithm's performance in terms of relevance and efficiency.*

*No changes are planned based on this comment.*

Section 4

- They conceptually compare the proposed method with other approaches for wind farm layout optimisation, such as genetic algorithms (which require a high number of goal function evaluations). However, they do not include, for instance, a quantitative comparison about computational times or wind farm efficiency gains.

  *We agree that quantitative comparisons are valuable to establish the performance of our approach relative to others. We are aware that more comparisons would be better (cf. page 31 lines 18–19), but feel what we currently have is sufficient for this paper, given the effort required for such comparisons (other group's implementations are hard to get and run, which is an important justification for the organisation of joint case studies).*

  *We have participated in the IEA Wind Task 37's Case Studies on wind farm layout optimization. The results of the first set of case Studies are referred to in Section 4.3.2 (Baker et al., 2019a), which quantitatively support our claims in terms of computational efficiency and performance; our Fig. 13 on page 24 summarizes the performance part. The results for the second set of Case Studies are the subject of a joint paper-in-progress by the participants (not yet citable); it reinforces the impression obtained of the first set.*

*We plan to add a figure mirroring Baker et al.'s Figure 5 next to our Figure 13, as this will make the efficiency/speed claim more explicitly quantitatively justified.*

*Only in case the editor feels this is needed, will we add an extra subsection to Section 4.3 grouping Figure 13, this new figure, and our argumentation. (We prefer not to, because the quantitative results are essentially restricted to the case study discussed in 4.3.2.)*

**RC2 (Andrew P. J. Stanley)**

Section 2

- I recommend changing the bullets on page 3 (lines 25-30). They look like minus signs.

  *We agree that bullets are less confusing and a more generally recognized way to start items in lists. However, the journal style uses dashes and there is nothing we as authors can do about it.*

  *We do not plan to take any action, as our experience is that the journal strictly follows its own style guide, which is a reasonable policy.*

- Page 7 line 6: should be "this value is the same for all turbines"?

  *Correct.*

  *We will fix this in the revision.*

- Page 7 line 17: clarify that they are equivalent because you assume a fixed number of turbines.

  *That would indeed clarify this.*

  *We will add this in the revision.*

- 4. Reading through the manuscript, my impression is that the objective function really doesn't matter that much, as the pseudo-gradient direction is determined spatially and the step size is not really the true gradient. Have you tried a similar method with just spatial information? You would just need one or two tuning parameters that would be used to determine step size as a function of distance, but you wouldn't need a wake model. Has this been explored at all? (note, I don't necessarily think this needs to be fully explored or resolved for this publication, but I do think others might have a similar question so it might be worth addressing in the text)

*It is true that the pseudo-gradient (step size) is not the true gradient, but the heuristic relies on the hypothesis that it is a decent proxy for the true gradient. As we can see in Figures 9 and 10 on page 21, the pseudo-gradients clearly are sensitive to the wind rose and not just inter-turbine distances. A situation where the wind rose is uniform could perhaps be handled with spatial information only. However, in such a situation or when restricting to spatial information only, a densest circle packing (hexagonal grid) would be a natural solution. This is actually what we use for our initial layout and it often already gives respectable results, given its simplicity. Nevertheless, as shown in Section 4.2, a wind-rose and wake-model aware pseudo-gradient-based algorithm easily improves upon it.*

*As an approximation to the spatial-information-only idea, one could consider using extremely simple wake models to speed up the algorithm, but that would just be a proportionality change in computational complexity and already fits into the setup of our paper, where the wake model specifics are intentionally de-emphasized.*

*We plan to expand Section 3.1, adding text that makes a smoother transition between the*

*definition of the objective function and the definition of pseudo-gradients. It will make the objective function proxy nature of pseudo-gradients explicit and also allow us to mention how objective function information related to other wind farm design variables, such as substructure cost, can be included.*

- 1. In the overview of the pseudo-gradients (starting with section 3.1), it is not intuitive to me why you chose three pseudo-gradient definitions to apply to the waked turbine, and only one to the waking turbine. It seems like the three pseudo-gradients would overdefine the downwind turbine and limit the upwind. Why not use two for each?

*We address the choice of the pseudo-gradient types discussed on page 9, lines 13-29. In brief, we there say that our current selection of pseudo-gradient types cover the qualitatively different possibilities we identified. Choosing 3 pseudo-gradients for the downstream turbine makes it easier to explain the differences in orientation and projection, since these vectors have the same starting point.*

*Part of your question may be related to the use of pseudo-gradients (we are not sure). These sections only discuss how the pseudo-gradients are defined. It is not meant to imply that the presented ones will be used as illustrated.*

*We plan to add a sentence at the end of Section 3.1 saying that while we present the different types of pseudo-gradient together in the text and figures, this does not mean they will be used together.*

- Page 8 line 1: "away" and "back" were initially unclear to me. It would be helpful to clarify, maybe with relative terms? "Further downstream" and "further upstream" could potentially work.

*The terms 'push-away' and 'push-back' are defined on page 8 lines 20–21 and page 9 lines 1–2. Figure 1 on page 8 makes it already more explicit what we mean. We feel that "to move the waked turbine away from the waking turbine"*

*provides sufficient explanation for 'push-away'. A bit more explanation could be added for 'push-back'.*

*In any case, adding upstream and downstream to the mix would cause confusion, as the direction of movement is not aligned with the downstream-to-upstream direction. Originally, in our code, we used 'push-down' instead of push-away, but because this triggers the wrong intuition, we changed to the more generic term 'push-away'. We could have used 'push-front', but that sounds strange to us. In the end, this is defined terminology, and the mathematical definition is unambiguous.*

*We plan to add a few more word of explanation for 'push-back'.*

- 5. The magnitude of the cross-stream pseudo-gradient seems arbitrary. Using a projection of one of the other pseudo-gradients (if I understood correctly) means the magnitude would always be relatively small. Often though the cross-stream gradients would be much higher than the stream wise ones (moving side to side gives much faster gains than moving further apart). It seems your formulation prioritizes that opposite. Can you explain or justify this?

*When looking at the figures in Section 3, it is indeed clear that in terms of wake loss units, projected pseudo-gradient types will be smaller. The projection also makes their magnitude relative to other types arbitrary. However, the relative magnitudes of the cross-type pseudo-gradient vectors for different turbines is not arbitrary and that is what is exploited. Namely, as we state (e.g., on page 8 lines 10–11) the wake loss unit-based vectors need to be translated to a spatial one using some proportionality constant. That proportionality constant does not need to be the same for all types, but per type it is the same for all turbines. Hence, the (per type) differences in magnitude do matter.*

*Actually, what we do in the algorithms, is normalize the pseudo-gradients (cf. Algorithm 2 step 3) and select a step size (cf., e.g., Algorithm 1 step 4), so only*

*relative magnitude information is retained from the original vectors when optimizing. Furthermore, in Algorithm 7 the chosen adaptive step size evolves separately for the different pseudo-gradient types, so that each type has the opportunity to 'compete' with the others, even if a different step size is needed for that. Therefore, our algorithmic design consciously removes biases in the original pseudo-gradients and even (based on wake loss reduction effect) biases present after normalization.*

*We plan to mention in Section 3 that the relative difference in magnitudes between the types will be removed in the optimization algorithms. We also plan to add sentences to Section 4.2 to make the reasons for normalization and per-type adaptive step sizes (more) explicit.*

Section 4

- 2. This may be related to the first question, so they may both be clarified in one step. When I first read through I understood (or maybe just assumed) that the pseudo-gradients would be combined to determine the step size and direction during optimization. However, when reading through the results it seemed like you chose the one pseudo-gradient type that achieved the best gain, and only took a step according to that pseudo-gradient. Which is correct? If you do only use one pseudo-gradient to determine the step, why not combine them to achieve better/faster convergence?

*The assumption that the pseudo-gradients would (always) be combined to determine the steps is not correct. They can be, but this is not necessary. In effect, choices how to use one or more different pseudo-gradients are part of the algorithm development process. In the proof-of-concept algorithms, only one pseudo-gradient type is used to define a step, not a linear combination of different pseudo-gradient types.*

*It is entirely possible to use linear combinations of different pseudo-gradient types. This is mentioned in the result discussion, on page 31, lines 30-32, and the conclusions, on page 33, line 22. We also implemented this in a simple way in the code to explore whether it could help convergence or performance. The results of the initial investigation were not promising enough to put more effort in it at this stage, but not definite enough to conclude this avenue of research could not be fruitful.*

*The change planned due to comment 1 will reduce the risk of of incorrect assumptions. No further changes are planned due to this comment, as combinations are mentioned already.*

- 6. It seems like a main outcome of this work is to be able to quickly and efficiently perform wind farm layout optimization that finds an improved layout, but much faster than other methods. However, you haven't provided any of that comparison. You probably don't need it for everything, but for one or two of the cases can you provide comparisons of optimizer performance and computational expense with some other methods? It seems like from a performance perspective, optimizing with exact analytic gradients would be just as fast as your proposed method, and achieve a better result. That would certainly not discount using pseudo-gradients to perform layout optimization, as often exact analytic gradients are difficult and time consuming to derive, but it would help frame it better.

*Regarding the comparison with other methods, please see our response to RC1's Section 4 comment.*

*Regarding the comparison with exact analytic gradients, as you state, they are difficult and time consuming to derive. Whether they can be as fast, we do not know and we do not have the expertise to set up a study doing a fair comparison (that is something a joint benchmark/case study can investigate). Pseudo-gradients are numerical and therefore, once a pseudo-gradient has been defined for an aspect*

*(wake, substructure) of the optimization problem, they apply to many models for that aspect.*

*Currently, we mention heuristic algorithms in the introduction and sketch why they are time consuming. We plan, to help the framing as you suggest, to also mention analytic and numerical gradient-based methods there in the same way.*

- 3. Not an issue, more a curiosity that other people might have as well. Have you considered including the constraint information in the pseudo-gradients as well, as to not need to force feasible solutions and potentially gain more exploration?

*As alluded to on page 32, lines 23–31, there have been efforts to include constraint information as a separate vector (similar to a tangent-bug algorithm). This has proved successful in navigating around wrecks and other small obstacles, as well as along concave site boundaries. However, as with any optimization algorithm that relies on shifting the nodes around, there remain difficulties with disjoint sites or sites that are otherwise crossed by exclusion zones (e.g., pipeline routes).*

*There are no plans to change the paper based on this comment, as the developments in this area are either proprietary or too immature.*

---

## Author Response (AR1)

**List of revisions**

Erik Quaeghebeur, René Bos, and Michiel B. Zaaijer

April 6, 2021

This list of revisisons is structured as reactions per reviewer and then follows the general sectioning structure of the paper (so we may reorder some comments), but first the general comments are treated. The format is as follows:

- The reviewer comment is shown in an upright font.

  *Our response follows in blue and slanted font.*

  *The implemented changes are described in red and slanted smaller font.*

**RC1 (Anonymous)**

**General comments**

- [...] in my opinion the paper should be mainly reviewed by experts in pure abstract aspects of optimization algorithms who would determine if this approach is really new, relevant and efficient as the authors claims.

  *We feel a review by an expert in the general field provides value, even if they cannot judge algorithmic details or sub-field relevance. We agree that scrutiny of experts on wind farm layout optimization is desirable and this is provided by the second reviewer. We think it is up to the editor to decide whether sufficient expertise is covered by both reviews. Outside of the review process for this paper, we have participated using the proposed algorithm in IEA Wind Task 37's Case Studies on wind farm layout optimization. This is not the same as a review, but does provide a check of the algorithm's performance in terms of relevance and efficiency.*

  *No changes have been made based on this comment.*

**Section 4**

- They conceptually compare the proposed method with other approaches for wind farm layout optimisation, such as genetic algorithms (which require a high number of goal function evaluations). However, they do not include, for instance, a quantitative comparison about computational times or wind farm efficiency gains.

  *We agree that quantitative comparisons are valuable to establish the performance of our approach relative to others. We are aware that more comparisons would be better (cf. page 31 lines 18–19), but feel what we currently have is sufficient for this paper, given the effort required for such comparisons (other group's implementations are hard to get and run, which is an important justification for the organisation of joint case studies).*

*We have participated in the IEA Wind Task 37's Case Studies on wind farm layout optimization. The results of the first set of case Studies are referred to in Section 4.3.2 (Baker et al., 2019a), which quantitatively support our claims in terms of computational efficiency and performance; our Fig. 13 on page 24 summarizes the performance part. The results for the second set of Case Studies are the subject of a joint paper-in-progress by the participants (not yet citable); it reinforces the impression obtained of the first set.*

*We have added a figure using data from the IEA Wind Task 37 Case Study 1 64-turbine scenario (new Figure 14). It plots wake loss percentage against the number of model calls. The latter quantity is a decent indicator of algorithm efficiency (more so than running time, which depends on implementation and hardware). It very explicitly shows the efficiency of pseudo-gradient-based algorithms.*

**RC2 (Andrew P. J. Stanley)**

**Section 2**

- I recommend changing the bullets on page 3 (lines 25-30). They look like minus signs.

  *We agree that bullets are less confusing and a more generally recognized way to start items in lists. However, the journal style uses dashes and there is nothing we as authors can do about it.*

  *No changes have been made based on this comment. We continue following the journal's prescribed style.*

- Page 7 line 6: should be "this value is the same for all turbines"?

  *Correct.*

  *This has been fixed.*

- Page 7 line 17: clarify that they are equivalent because you assume a fixed number of turbines.

  *That would indeed clarify this.*

  *We have added "Because $|\mathcal{T}|$ is fixed, ".*

**Section 3**

- 4. Reading through the manuscript, my impression is that the objective function really doesn't matter that much, as the pseudo-gradient direction is determined spatially and the step size is not really the true gradient. Have you tried a similar method with just spatial information? You would just need one or two tuning parameters that would be used to determine step size as a function of distance, but you wouldn't need a wake model. Has this been explored at all? (note, I don't necessarily think this needs to be fully explored or resolved for this publication, but I do think others might have a similar question so it might be worth addressing in the text)

  *It is true that the pseudo-gradient (step size) is not the true gradient, but the heuristic relies on the hypothesis that it is a decent proxy for the true gradient. As we can see in Figures 9 and 10 on page 21, the pseudo-gradients clearly are sensitive to the wind rose and not just inter-turbine distances. A situation where the wind rose is uniform could perhaps be handled with spatial information only. However, in such a situation or when restricting to spatial*

*information only, a densest circle packing (hexagonal grid) would be a natural solution. This is actually what we use for our initial layout and it often already gives respectable results, given its simplicity. Nevertheless, as shown in Section 4.2, a wind-rose and wake-model aware pseudo-gradient-based algorithm easily improves upon it.*

*As an approximation to the spatial-information-only idea, one could consider using extremely simple wake models to speed up the algorithm, but that would just be a proportionality change in computational complexity and already fits into the setup of our paper, where the wake model specifics are intentionally de-emphasized.*

We expanded Section 3.1 by prepending four new paragraphs, which create a smoother transition between the definition of the objective function and the definition of pseudo-gradients. It makes the objective function proxy nature of pseudo-gradients explicit and also makes it possible for us to mention how objective function information related to other wind farm design variables, such as substructure cost, could be included.

- 1. In the overview of the pseudo-gradients (starting with section 3.1), it is not intuitive to me why you chose three pseudo-gradient definitions to apply to the waked turbine, and only one to the waking turbine. It seems like the three pseudo-gradients would overdefine the downwind turbine and limit the upwind. Why not use two for each?

  *We address the choice of the pseudo-gradient types discussed on page 9, lines 13-29. In brief, we there say that our current selection of pseudo-gradient types cover the qualitatively different possibilities we identified. Choosing 3 pseudo-gradients for the downstream turbine makes it easier to explain the differences in orientation and projection, since these vectors have the same starting point.*

  *Part of your question may be related to the use of pseudo-gradients (we are not sure). These sections only discuss how the pseudo-gradients are defined. It is not meant to imply that the presented ones will be used as illustrated.*

  We have added two sentences at the end of Section 3.1 to clarify that the pseudo-gradient types can be used individually and that their use will be discussed later in the paper.

- Page 8 line 1: "away" and "back" were initially unclear to me. It would be helpful to clarify, maybe with relative terms? "Further downstream" and "further upstream" could potentially work.

  *The terms 'push-away' and 'push-back' are defined on page 8 lines 20–21 and page 9 lines 1–2. Figure 1 on page 8 makes it already more explicit what we mean. We feel that "to move the waked turbine away from the waking turbine" provides sufficient explanation for 'push-away'. A bit more explanation could be added for 'push-back'.*

  *In any case, adding upstream and downstream to the mix would cause confusion, as the direction of movement is not aligned with the downstream-to-upstream direction. Originally, in our code, we used 'push-down' instead of push-away, but because this triggers the wrong intuition, we changed to the more generic term 'push-away'. We could have used 'push-front', but that sounds strange to us. In the end, this is defined terminology, and the mathematical definition is unambiguous.*

  We now introduce push-back pseudo-gradients with "Instead of moving the waked turbine away, it is also possible to move the waking turbine back **relative to the waked turbine**.", where the phrase in bold is new.

- 5. The magnitude of the cross-stream pseudo-gradient seems arbitrary. Using a projection of one of the other pseudo-gradients (if I understood correctly) means the magnitude would

always be relatively small. Often though the cross-stream gradients would be much higher than the stream wise ones (moving side to side gives much faster gains than moving further apart). It seems your formulation prioritizes that opposite. Can you explain or justify this?

*When looking at the figures in Section 3, it is indeed clear that in terms of wake loss units, projected pseudo-gradient types will be smaller. The projection also makes their magnitude relative to other types arbitrary. However, the relative magnitudes of the cross-type pseudo-gradient vectors for different turbines is not arbitrary and that is what is exploited. Namely, as we state (e.g., on page 8 lines 10–11) the wake loss unit-based vectors need to be translated to a spatial one using some proportionality constant. That proportionality constant does not need to be the same for all types, but per type it is the same for all turbines. Hence, the (per type) differences in magnitude do matter.*

*Actually, what we do in the algorithms, is normalize the pseudo-gradients (cf. Algorithm 2 step 3) and select a step size (cf., e.g., Algorithm 1 step 4), so only relative magnitude information is retained from the original vectors when optimizing. Furthermore, in Algorithm 7 the chosen adaptive step size evolves separately for the different pseudo-gradient types, so that each type has the opportunity to 'compete' with the others, even if a different step size is needed for that. Therefore, our algorithmic design consciously removes biases in the original pseudo-gradients and even (based on wake loss reduction effect) biases present after normalization.*

*In Section 3.4, at the end of the discussion of push-cross pseudo-gradients, we have added the sentence "Such somewhat arbitrary differences in resulting magnitude between pseudo-gradient types can be normalized away before using them in optimization algorithms and should therefore not be a cause for concern." We have also added a paragraph at the end of Section 4.2 to make the presence and effects of normalization and per-type adaptive step sizes (more) explicit.*

**Section 4**

- 2. This may be related to the first question, so they may both be clarified in one step. When I first read through I understood (or maybe just assumed) that the pseudo-gradients would be combined to determine the step size and direction during optimization. However, when reading through the results it seemed like you chose the one pseudo-gradient type that achieved the best gain, and only took a step according to that pseudo-gradient. Which is correct? If you do only use one pseudo-gradient to determine the step, why not combine them to achieve better/faster convergence?

*The assumption that the pseudo-gradients would (always) be combined to determine the steps is not correct. They can be, but this is not necessary. In effect, choices how to use one or more different pseudo-gradients are part of the algorithm development process. In the proof-of-concept algorithms, only one pseudo-gradient type is used to define a step, not a linear combination of different pseudo-gradient types.*

*It is entirely possible to use linear combinations of different pseudo-gradient types. This is mentioned in the result discussion, on page 31, lines 30-32, and the conclusions, on page 33, line 22. We also implemented this in a simple way in the code to explore whether it could help convergence or performance. The results of the initial investigation were not promising enough to put more effort in it at this stage, but not definite enough to conclude this avenue of research could not be fruitful.*

*Because the change implemented due to comment 1 reduces the risk of incorrect assumptions, we have made no further changes due to this comment, as combinations are mentioned already.*

- 6. It seems like a main outcome of this work is to be able to quickly and efficiently perform wind farm layout optimization that finds an improved layout, but much faster than other methods. However, you haven't provided any of that comparison. You probably don't need it for everything, but for one or two of the cases can you provide comparisons of optimizer performance and computational expense with some other methods? It seems like from a performance perspective, optimizing with exact analytic gradients would be just as fast as your proposed method, and achieve a better result. That would certainly not discount using pseudo-gradients to perform layout optimization, as often exact analytic gradients are difficult and time consuming to derive, but it would help frame it better.

  *Regarding the comparison with other methods, please see our response to RC1's Section 4 comment.*

  *Regarding the comparison with exact analytic gradients, as you state, they are difficult and time consuming to derive. Whether they can be as fast, we do not know and we do not have the expertise to set up a study doing a fair comparison (that is something a joint benchmark/case study can investigate). Pseudo-gradients are numerical and therefore, once a pseudo-gradient has been defined for an aspect (wake, substructure) of the optimization problem, they apply to many models for that aspect.*

  *To, as you suggest, help the framing, we have expanded the first sentence of the last paragraph of Section 1.1 into a paragraph. It mentions the downsides of meta-heuristics (kept), numerical gradient-based (new) and analytical gradient-based (new).*

- 3. Not an issue, more a curiosity that other people might have as well. Have you considered including the constraint information in the pseudo-gradients as well, as to not need to force feasible solutions and potentially gain more exploration?

  *As alluded to on page 32, lines 23–31, there have been efforts to include constraint information as a separate vector (similar to a tangent-bug algorithm). This has proved successful in navigating around wrecks and other small obstacles, as well as along concave site boundaries. However, as with any optimization algorithm that relies on shifting the nodes around, there remain difficulties with disjoint sites or sites that are otherwise crossed by exclusion zones (e.g., pipeline routes).*

  *We have not changed the paper based on this comment, as the developments in this area are either proprietary or too immature.*